# Constructing Orthogonal Convolutions in an Explicit Manner

**Tan Yu, Jun Li, Yunfeng Cai, Ping Li**

Cognitive Computing Lab
Baidu Research
10900 NE 8th St. Bellevue, Washington 98004, USA
`{tanyu01,lijun12,caiyunfeng,liping11}@baidu.com`

## Abstract

Convolutions with orthogonal input-output Jacobian matrix, i.e., orthogonal convolution, have recently attracted substantial attention. A convolution layer with an orthogonal Jacobian matrix is 1-Lipschitz in the 2-norm, making the output robust to the perturbation in input. Meanwhile, an orthogonal Jacobian matrix preserves the gradient norm in back-propagation, which is critical for stable training deep networks. Nevertheless, existing orthogonal convolutions are burdened by high computational costs for preserving orthogonality. In this work, we exploit the relation between the singular values of the convolution layer's Jacobian and the structure of the convolution kernel. To achieve orthogonality, we explicitly construct the convolution kernel for enforcing all singular values of the convolution layer's Jacobian to be 1s. After training, the explicitly constructed orthogonal (ECO) convolutions are constructed only once, and their weights are stored. Then, in evaluation, we only need to load the stored weights of the trained ECO convolution, and the computational cost of ECO convolution is the same as the standard dilated convolution. It is more efficient than the recent state-of-the-art approach, skew orthogonal convolution (SOC) in evaluation. Experiments on CIFAR-10 and CIFAR-100 demonstrate that the proposed ECO convolution is faster than SOC in evaluation while leading to competitive standard and certified robust accuracies.

## 1 Introduction

A layer with an orthogonal input-output Jacobian matrix is 1-Lipschitz in the 2-norm, robust to the perturbation in input. Meanwhile, it preserves the gradient norm in back-propagating the gradient, which effectively overcomes the gradient explosion and attenuation issues in training deep neural networks. In the past years, many studies have shown that exploiting orthogonality of the Jacobian matrix of layers in neural networks can achieve provable robustness to adversarial attacks (Li et al., 2019), stabler and faster training (Arjovsky et al., 2016; Xiao et al., 2018), and improved generalization (Cogswell et al., 2016; Bansal et al., 2018; Sedghi et al., 2019).

In a fully-connected layer $\mathbf{y} = \mathbf{W}\mathbf{x}$ where $\mathbf{W} \in \mathbb{R}^{c_{out} \times c_{in}}$ is the weight matrix, the layer's input-output Jacobian matrix $\mathbf{J} = \frac{\partial \mathbf{y}}{\partial \mathbf{x}}$ is just $\mathbf{W}$. Thus, preserving the orthogonality of $\mathbf{J}$ can be accomplished through imposing the orthogonal constraint on $\mathbf{W}$, which has been extensively studied in previous works (Mhammedi et al., 2017; Cissé et al., 2017; Anil et al., 2019). In contrast, in a convolution layer, the Jacobian matrix is no longer the weight matrix (convolution kernel). Instead, it is the circulant matrix composed of convolution kernel (Karami et al., 2019; Sedghi et al., 2019). Thus, generally, simply constructing an orthogonal convolution kernel cannot achieve an orthogonal convolution. Achieving an orthogonal Jacobian matrix in the convolution layer is more challenging than that in a fully-connected layer. It is plausibly straightforward to expand the convolution kernel to the doubly block circulant Jacobian matrix and impose the orthogonal constraint. But it is difficult to construct a Jacobian matrix that is both doubly block-circulant and orthogonal.

Block Convolutional Orthogonal parameterization (BCOP) (Li et al., 2019) is one of the pioneering works for constructing the orthogonal convolution neural networks. It adopts the construction algorithm (Xiao et al., 2018) which decomposes a 2-D convolution into a stack of 1-D convolutions and a channel-wise orthogonal transformation. Trockman & Kolter (2021) maps the convolution kernel

and the feature tensor into the frequency domain using Fast Fourier Transform (FFT), and achieves the orthogonality of the Jacobian matrix through Cayley transform on the weight matrix in the frequency domain. They devise the convolution kernel of the same size as the input feature map, which takes more parameters than standard convolution layers with a local reception field. Meanwhile, FFT maps the real-value feature map and the weight matrix into matrices of complex values, increasing the computational cost to 4 times its counterpart in the real-value domain. Meanwhile, the Cayley transform requires computing the matrix inverse, which is not friendly for GPU computation. Recently, Skew Orthogonal Convolutions (SOC) (Singla & Feizi, 2021b) devises skew-symmetric filter and exploits matrix exponential (Hoogeboom et al., 2020) to attain the orthogonality of the Jacobian matrix. But SOC is slow in evaluation since it needs to apply a convolution filter multiple times sequentially on the feature map to obtain the Taylor expansion. Observing the efficiency limitations of the existing methods, in this work, we propose an explicitly constructed orthogonal (ECO) convolution, which is fast in both training and evaluation.

It is well known that the Jacobian matrix of a layer is orthogonal if and only if each of its singular values is 1. Thus we convert the problem of ensuring the orthogonality of the Jacobian matrix into making every singular value as 1. Based on the relation between the singular values of the Jacobian matrix for the convolution layer and the weight matrix of the convolution layer discovered by Sedghi et al. (2019), we construct the convolution kernel so that it ensures every singular value of the Jacobian matrix to be 1. Compared with the recent state-of-the-art method, SOC (Singla & Feizi, 2021b) implicitly approximating orthogonal convolution by multiple times convolution operations, ours explicitly builds the orthogonal convolution. Thus, we can directly deploy the constructed orthogonal convolution in evaluation, taking the same computational cost as the standard convolution. It is more efficient than SOC with multiple times convolution operations in evaluation. Experiments on CIFAR10 and CIFAR100 show that, in evaluation, taking less time, ours achieves competitive standard and robust accuracy compared with SOC.

## 2   RELATED WORK

**Weight orthogonalization.**   In a fully-connected layer, the Jacobian matrix is the weight matrix itself. Thus many efforts are devoted to orthogonalizing the weights. Early works exploit an orthogonal weight initialization to speed up the training (Saxe et al., 2014; Pennington et al., 2017) and build deep neural networks (Xiao et al., 2018). Recently, more efforts are devoted to exploiting the orthogonality in training. Some approaches propose "soft" constraints on the weights. For example, Bansal et al. (2018); Xiong et al. (2016) introduce mutual coherence and spectral restricted isometry as a regularization on the weight matrix. Parseval Networks (Cissé et al., 2017) adapts a regularizer to encourage the orthogonality of the weight matrices. However, these methods cannot guarantee the exact orthogonality of the weight matrix. Some approaches orthogonalize features during the forward pass. For example, Huang et al. (2018b) extends Batch Normalization (Ioffe & Szegedy, 2015) with ZCA; Huang et al. (2018a) solves a suboptimization problem to decorrelate features. These methods cannot avoid computationally expensive operations like SVD. Other approaches enforce orthogonality by the Riemannian optimization on the Stiefel manifold. For example, Casado & Martínez-Rubio (2019) uses Pade approximation as an alternative to matrix exponential mapping to update the weight matrix on the Stiefel manifold. Li et al. (2020) uses an iterative Cayley transform to enforce weight matrix orthonormal. Anil et al. (2019) also attains orthogonal weights by orthogonal parameterization based on Björck orthonormalization (Björck & Bowie, 1971). Lastly, some approaches incorporate orthogonal constraint into the network architecture. For example, Mhammedi et al. (2017) builds orthogonal layers in RNNs with householder reflections.

**Orthogonal convolution.**   In a convolution layer, the Jacobian matrix is no longer the weight matrix. Thus, the above-mentioned weight orthogonalization methods cannot be trivially applied in convolution layers. Wang et al. (2020); Qi et al. (2020) encourage the orthogonality of the Jacobian matrix of convolution layers through a regularizer, but they cannot achieve the strict orthogonality and cannot attain the provable robustness. Block Convolutional Orthogonal parameterization (BCOP) (Li et al., 2019) is a pioneering work for enforcing the orthogonality of the Jacobian matrix of a convolution layer. It conducts parameterization of orthogonal convolutions by adapting the construction algorithm proposed by Xiao et al. (2018). But BCOP is slow in training. Trockman & Kolter (2021) applies the Cayley transform to a skew-symmetric convolution weight in the Fourier domain so that

the convolution recovered from the Fourier domain has an orthogonal Jacobian matrix. Though it achieves higher efficiency in some applications than BCOP, it is still significantly more costly than standard convolutional layers. Skew Orthogonal Convolution (Singla & Feizi, 2021b) achieves orthogonal convolution through Taylor expansion of the matrix exponential. It is faster than BCOP in training but slower in evaluation. In contrast, our method is efficient in both training and evaluation. Recently, Su et al. (2021) achieves orthogonal convolutions via paraunitary systems.

## 3 PRELIMINARY

**Notation.** For a matrix $\mathbf{M}$, $\mathbf{M}[i, j]$ denotes the element in the $i$-th row and the $j$-th column, $\mathbf{M}[i, :]$ denotes the $i$-th row and $\mathbf{M}[:, j]$ denotes the $j$-th column. We use the similar notation for indexing tensors. For $n \in \mathbb{N}$, $[n]$ denotes the set of $n$ numbers from 0 to $n - 1$, that is, $\{0, 1, \cdots, n - 1\}$. We denote the function to obtain the singular values by $\sigma(\cdot)$. Let $\text{vec}(\cdot)$ denotes the function which unfolds a matrix or higher-order tensor into a vector. Let $\omega_k = \exp(2\pi i/k)$ where $i = \sqrt{-1}$. Let $\mathbf{F}_k \in \mathbb{R}^{k \times k}$ be matrix for the discrete Fourier transform (DFT) for $k$-element vectors and each entry of $\mathbf{F}_k$ is defined as $\mathbf{F}_k[p, q] = \omega_k^{pq}$.

### 3.1 RELATION BETWEEN THE JACOBIAN MATRIX AND THE WEIGHT MATRIX

We denote the input tensor of a convolution layer by $\mathcal{X} \in \mathbb{R}^{c \times n \times n}$, where $n$ is the spatial size, and $c$ is the number of input channels. We denote the convolution operation on $\mathcal{X}$ by $\text{conv}_{\hat{\mathcal{W}}}(\mathcal{X})$, where $\hat{\mathcal{W}} \in \mathbb{R}^{k \times k \times c \times c}$ size where $k$ is the size of the reception field and $k$ is normally smaller than size of the input image, $n$. In this definition, we set the number of output channels identical to that of input channels. Meanwhile, we set the convolution stride as 1 and adapt the cyclic padding to make the output of the same size as the input.

We denote the output tensor by $\mathcal{Y} \in \mathbb{R}^{c \times n \times n}$ and each element of $\mathcal{Y}$ is obtained by

$$\forall l, s, t \in [c] \times [n] \times [n], \quad \mathcal{Y}[l, s, t] = \sum_{r \in [c]} \sum_{p \in [k]} \sum_{q \in [k]} \mathcal{X}[l + r, s + p, t + q] \hat{\mathcal{W}}[p, q, l, r]. \quad (1)$$

Since the convolution operation is a linear transformation, there exists a linear transformation $\mathbf{M} \in \mathbb{R}^{n^2 c \times n^2 c}$ which satisfies:

$$\mathcal{Y} = \text{conv}_{\hat{\mathcal{W}}}(\mathcal{X}) \Leftrightarrow \text{vec}(\mathcal{Y}) = \mathbf{M}\text{vec}(\mathcal{X}). \quad (2)$$

To simplify illustration, $\hat{\mathcal{W}} \in \mathbb{R}^{k \times k \times c \times c}$ is normally expanded into $\mathcal{W} \in \mathbb{R}^{n \times n \times c \times c}$ through zero-padding. We term $\mathcal{W}$ as the expanded convolution kernel. In $\mathcal{W}$, only $k^2 c^2$ elements are non-zero. Based on the above definition, we review the following theorem in Sedghi et al. (2019):

**Theorem 1** (see Sedghi et al. (2019) Section 2.2). *For any expanded convolution kernel* $\mathcal{W} \in \mathbb{R}^{n \times n \times c \times c}$, *let* $\mathbf{M}$ *is the matrix encoding the linear transformation of a convolution with* $\mathcal{W}$. *For each* $p, q \in [n] \times [n]$, *let* $\mathbf{P}^{(p,q)}$ *be the* $c \times c$ *matrix computed by*

$$\forall s, t \in [c] \times [c], \quad \mathbf{P}^{(p,q)}[s, t] = (\mathbf{F}_n^\top \mathcal{W}[:, :, s, t] \mathbf{F}_n)[p, q]. \quad (3)$$

*Then*

$$\sigma(\mathbf{M}) = \bigcup_{p \in [n], q \in [n]} \sigma(\mathbf{P}^{(p,q)}). \quad (4)$$

The above theorem gives the relation between the singular values of the transformation matrix $\mathbf{M}$ and the expanded convolution $\mathcal{W}$.

**Theorem 2.** *For any* $\mathcal{W} \in \mathbb{R}^{n \times n \times c \times c}$, *the Jacobian matrix of the convolution layer,* $\mathbf{J}$, *is orthogonal if and only if* $\forall p, q \in [n] \times [n]$, *each singular value of the matrix* $\mathbf{P}^{(p,q)}$ *is* 1.

**Proof.** A real matrix is orthogonal if and only if each singular value is 1. Meanwhile, the Jacobian matrix $\mathbf{J} = \frac{\partial \text{vec}(\mathcal{Y})}{\partial \text{vec}(\mathcal{X})} = \mathbf{M}$. Based on Theorem 1, each singular value of $\mathbf{M}$ is 1 if and only if $\forall p, q \in [n] \times [n]$, each singular value of the matrix $\mathbf{P}^{(p,q)}$ is 1, thus we complete the proof.

## 3.2 Matrix Orthogonalization

**Carley transform.** Orthogonal parameterization has been extensively studied in the previous works. Many works (Absil et al., 2008; Helfrich et al., 2018; Maduranga et al., 2019) utilize Cayley transform on skew-symmetric matrices ($\mathbf{A}^\top = $ -$\mathbf{A}$) to obtain the orthogonal weight matrices. Cayley transform is a bijection between skew-symmetric matrices and orthogonal matrices without $-1$ eigenvalues. Specifically, it maps a skew-symmetric matrix $\mathbf{A}$ to an orthogonal matrix $\mathbf{Q}$ by

$$\mathbf{Q} = (\mathbf{I} - \mathbf{A})(\mathbf{I} + \mathbf{A})^{-1}. \tag{5}$$

Li et al. (2020) proposes an iterative approximation of the Cayley transform for orthogonally-constrained optimizers and achieves higher speed for training CNNs and RNNs.

**Matrix exponential.** Casado & Martínez-Rubio (2019) exploits the matrix exponential of the skew-symmetric matrix for orthogonality. Specifically, if $\mathbf{A}$ is a skew-symmetric matrix, $\exp(\mathbf{A})$ is orthogonal. SOC (Singla & Feizi, 2021b) approximates $\exp(\mathbf{A})$ through Taylor series $\exp(\mathbf{A}) \approx \mathbf{S}_T(\mathbf{A}) = \sum_{i=0}^{T} \frac{\mathbf{A}^i}{i!}$, and derives a bound on the approximation error:

$$\|\exp(\mathbf{A}) - \mathbf{S}_T(\mathbf{A})\|_2 \leq \frac{\|\mathbf{A}\|_2^T}{T!}, \tag{6}$$

where $\|\cdot\|_2$ is the spectral norm, the maximum singular value of a matrix. Since the approximation error is bounded by $\|\mathbf{A}\|_2$, SOC adopts the spectral normalization on $\mathbf{A}$ to make $\|\mathbf{A}\|_2$ small.

**Orthogonality of convolution.** In a fully-connected layer, its Jacobian matrix is the transpose of the weight matrix. Thus we can achieve orthogonality of the Jacobian matrix through orthogonal parameterization on the weight matrix through common orthogonal parameterization methods such as Cayley transform or matrix exponential of the skew-symmetric matrix. However, in a convolution layer, the Jacobian matrix is no longer the transpose of the weight matrix; hence we cannot just apply orthogonal parameterization on the weight matrix to achieve orthogonality of the Jacobian matrix.

## 3.3 Lipschitzness and Provably Robustness

**Lipschitz constant.** A function $f : \mathbb{R}^m \to \mathbb{R}^n$ is $L$-Lipschitz under the $\ell_2$ norm iff $\|f(\mathbf{x}) - f(\mathbf{y})\|_2 \leq L\|\mathbf{x} - \mathbf{y}\|_2, \forall (\mathbf{x}, \mathbf{y}) \in \mathbb{R}^m \times \mathbb{R}^m$, where $L$ is a positive constant. We define the smallest value of $L$ satisfying $L$-Lipschitz for $f$ as the Lipschitz constant of $f$, denoted by $\mathrm{Lip}(f)$. Given two Lipschitz continuous functions $f$ and $g$, the Lipschitz constant of the a composition of them, $\mathrm{Lip}(f \circ g)$, is upper-bounded by the product of the Lipschitz constants of $f$ and $g$ (Weaver, 2018):

$$\mathrm{Lip}(f \circ g) \leq \mathrm{Lip}(f)\mathrm{Lip}(g). \tag{7}$$

Thus, given a network $n(\cdot)$ consisting of $L$ layers, if the Lipschitz constant of each layer is 1, the Lipschitz constant of the whole network, $\mathrm{Lip}(n)$ is less than 1.

**Provably robustness.** Considering a neural network $n(\cdot)$ for $c$-class classification. Given an input $\mathbf{x}$ with the ground-truth label $y \in [c]$, the network generates a $c$-dimensional prediction vector $n(\mathbf{x}) = [s_1, \cdots, s_c]$ where the $i$-th item in $n(\mathbf{x})$ denotes the confidence score for the class $i$. The margin of prediction for $\mathbf{x}$ by $n(\cdot)$ is defined as

$$\mathcal{M}_n(\mathbf{x}) = \max(0, s_y - \max_{i \neq y} s_i). \tag{8}$$

As proved by Tsuzuku et al. (2018) and Li et al. (2019), for a network $n(\cdot)$ with a Lipschitz constant of $L$, $\mathbf{x}$ is provably robustly classified by $n(\cdot)$ under perturbation with a $\ell_2$-norm of $\frac{\mathcal{M}_n(\mathbf{x})}{\sqrt{2}L}$, that is,

$$\arg\max_i n(\mathbf{x} + \boldsymbol{\delta}) = y, \ \forall \|\boldsymbol{\delta}\|_2 < \frac{\mathcal{M}_n(\mathbf{x})}{\sqrt{2}L}. \tag{9}$$

## 4 Method

Based on Theorem 2, to achieve an orthogonal Jacobian matrix $\mathbf{J}$ for a convolution layer, we need to ensure that, $\forall p, q \in [n] \times [n]$, each singular value of the matrix $\mathbf{P}^{(p,q)}$ is 1. When $\mathbf{P}^{(p,q)}$ is a real matrix, this condition is equivalent to the orthogonality of $\mathbf{P}^{(p,q)}$. To simplify the notation, we assemble the matrices $\mathbf{P}^{(p,q)}$ ($p, q \in [n] \times [n]$) into a tensor $\mathcal{P} \in \mathbb{R}^{n \times n \times c \times c}$ such that $\mathcal{P}[p, q, :, :] = \mathbf{P}^{(p,q)}$. Based on the definition, we rewrite the transformation from $\mathcal{W}$ to $\mathbf{P}^{(p,q)}$ in Eq. (3) into

$$\forall s, t \in [c] \times [c], \mathcal{P}[:, :, s, t] = \mathbf{F}_n^\top \mathcal{W}[:, :, s, t] \mathbf{F}_n. \tag{10}$$

Since $\mathbf{F}_n$ is the full-rank square matrix, the mapping from $\mathcal{W}$ to $\mathcal{P}$ is invertible. Thus, to achieve an orthogonal Jacobian matrix, we can directly construct orthogonal $\mathcal{P}[p, q, :, :]$ ($p, q \in [n] \times [n]$) using Carley transform or matrix exponential of skew-symmetric matrices (introduced in Section 3.2) to make each singular value of them identical to 1. Then the convolution kernel weight $\mathcal{W}$ can be determined uniquely from the $\mathcal{P}$ through the inverse transformation in Eq. (10). Since $\mathbf{F}_n$ is the Discrete Fourier Transform (DFT) matrix, $\mathbf{F}_n^\top \mathcal{W}[:, :, s, t]\mathbf{F}_n$ in Eq. (10) is equivalent to conducting a 2-dimensional DFT on $\mathcal{W}[:, :, s, t]$. Thus, we can recover $\mathcal{W}[:, :, s, t]$ through the inverse 2-dimensional DFT (IDFT$_{2D}$) on the constructed orthogonal matrix $\mathcal{P}[:, :, s, t]$. That is,

$$\mathcal{W}[:, :, s, t] = \text{IDFT}_{2D}(\mathcal{P}[:, :, s, t]), \ \ \forall s, t \in [c] \times [c]. \tag{11}$$

Based on Theorem 2, the Jacobian matrix of the convolution layer with the kernel $\mathcal{W}$ is orthogonal.

Nevertheless, the above approach has many drawbacks. First of all, for a generally constructed $\mathcal{P}$, the recovered expanded convolution kernel $\mathcal{W}$ computed based on the inverse 2-dimensional DFT might contain $n^2 c^2$ non-zeros elements. It significantly increases the number of parameters in the convolution layer from $k^2 c^2$ to $n^2 c^2$. Meanwhile, the convolution with kernel $\mathcal{W}$ has a global reception field, which is also slower than the original convolution kernel with a local reception field. Besides, the above approach needs to compute $n^2$ times matrix orthogonalization (Carley transform or matrix exponential) on $\mathcal{P}[p, q, :, :]$ ($p, q \in [n] \times [n]$). Considering the spatial size of the input feature tensor, $n$ and the input/out channel number $c$ are not a small value, $n^2$ times orthogonalization on $c \times c$ matrices might be computationally expensive.

Thus, to achieve a high efficient orthogonal convolution, we devise a 2-dimensional periodic $\mathcal{P}$. We set the period as $k$ and assume $n$ is divisible by $k$. Specifically, $\mathcal{P}$ satisfies

$$\mathcal{P}[p, q, :, :] = \mathcal{P}[p + i * k, q + j * k, :, :], \ \ \forall [p, q] \in [k] \times [k], [i, j] \in [n/k] \times [n/k]. \tag{12}$$

We define $\mathcal{P}_0 = \mathcal{P}[0{:}k, 0{:}k, :, :] \in \mathbb{R}^{k \times k \times c \times c}$. That is, $\mathcal{P}$ can be obtained by repeating $\mathcal{P}_0$ by $n/k \times n/k$ times in the first two dimensions. It has many benefits by the 2-dimensional periodic configuration. First of all, based on the property of DFT, the convolution kernel $\mathcal{W}[:, :, s, t]$ obtained from an inverse 2D DFT on the 2-dimensional periodic $\mathcal{P}[:, :, s, t]$ on contains only $k^2$ non-zero elements. That is, the number of non-zero elements in the recovered $\mathcal{W}$ is only $k^2 c^2$, which is the same as the number of elements in original convolution kernel $\hat{\mathcal{W}}$ with a local reception field of $k \times k$ size. Besides, the non-zero elements in $\mathcal{W}$ can be efficiently obtained. To be specific, we define $\mathcal{W}_0 \in \mathbb{R}^{k \times k \times c \times c}$ obtained from $\mathcal{W}_0[:, :, s, t] = \text{IDFT}_{2D}(\mathcal{P}_0[:, :, s, t])$. Based on the property of DFT (Nussbaumer, 1981), $\mathcal{W}$ and $\mathcal{W}_0$ have the following relation:

$$\mathcal{W}[i, j, :, :] = \begin{cases} \mathcal{W}_0[\frac{i*k}{n}, \frac{j*k}{n}, :, :], & \text{if } i\%\frac{n}{k} = 0 \text{ and } j\%\frac{n}{k} = 0, \\ \mathbf{0}^{c \times c}, & \text{otherwise.} \end{cases} \tag{13}$$

That is, we can obtain the non-zero elements of $\mathcal{W} \in \mathbb{R}^{n \times n}$ through computing $\mathcal{W}_0 \in \mathbb{R}^{k \times k}$, taking only $k^2$ times 2D FFT on $c \times c$ matrices. Since $\mathcal{P}$ is periodic along the first two dimensions, to achieve an orthogonal Jacobian matrix based on Theorem 2, we only need to explicitly construct $k^2$ orthogonal matrices $\mathcal{P}_0[p, q, :, :]$ ($(p, q) \in [k] \times [k]$) to ensure that $n^2$ matrices $\mathcal{P}[p, q, :, :]$ ($(p, q) \in [n] \times [n]$) are orthogonal. As the recovered $\mathcal{W}[i, j, :, :]$ is non-zero only if $i\%\frac{n}{k} = 0$ and $j\%\frac{n}{k} = 0$. This property corresponds to the dilated convolution (Yu & Koltun, 2016). Specifically, using the relation between $\mathcal{W}$ and $\mathcal{W}_0$ in Eq. (13), the following equation holds:

$$\text{Conv}_{\mathcal{W}}(\mathcal{X}) = \text{Conv}_{\mathcal{W}_0}(\mathcal{X}, \text{dilation} = \frac{n}{k}). \tag{14}$$

That is, we can achieve the orthogonal convolution through a dilated convolution with the kernel $\mathcal{W}_0$ and the dilation $n/k$. Based on above analysis, the process of achieving the orthogonal convolution is decomposed into three steps:

1. Construct the orthogonal matrices $\mathcal{P}_0[p, q, :, :], \forall(p, q) \in [k] \times [k]$.
2. Recover $\mathcal{W}_0[:, :, s, t]$ from $\mathcal{P}_0[:, :, s, t]$ using inverse 2D DFT, $\forall(s, t) \in [c] \times [c]$.
3. Conduct the dilated convolution using the recovered kernel $\mathcal{W}_0$ and the dilation $n/k$.

Below we introduce the details in these three steps individually.

**Orthogonal parameterization for $\mathcal{P}_0[p, q, :, :]$.** This step can be implemented by any matrix orthogonalization approach such as Carley Transform on skew-symmetric matrix or matrix exponential of the skew-symmetric matrix. By default, we adopt the Taylor expansion for the exponential of the skew-symmetric matrix proposed in SOC (Singla & Feizi, 2021b) in this step considering its simplicity and efficiency. To be specific, we start from a parameter tensor $\mathcal{P}_0 \in \mathbb{R}^{k \times k \times c \times c}$. $\forall (p, q) \in [k] \times [k]$, we compute

$$\mathcal{P}_0[p, q, :, :] \leftarrow \mathcal{P}_0[p, q, :, :] - \mathcal{P}_0[p, q, :, :]^\top,$$

$$\mathcal{P}_0[p, q, :, :] \leftarrow \sum_{i=0}^{T} \frac{\mathcal{P}_0[p, q, :, :]^i}{i!}. \tag{15}$$

Recall from Eq. (6) that the approximation error between $\exp(\mathbf{A})$ and the Taylor expansion $\mathbf{S}_k(\mathbf{A})$, $\|\exp(\mathbf{A}) - \mathbf{S}_k(\mathbf{A})\|_2$, is bounded by $\frac{\|A\|_2^k}{k!}$. That is, when $\|A\|_2 < 1$, it can achieve a good approximation. Thus, following SOC (Singla & Feizi, 2021b), we conduct spectral normalization on $\mathcal{P}_0[p, q, :, :]$ so that $\|\exp(\mathbf{P}_0[p, q, :, :]) - \sum_{i=0}^{T} \frac{\mathcal{P}_0[p, q, :, :]^i}{i!}\|_2 \leq \frac{\|\mathcal{P}_0[p, q, :, :]\|_2^k}{k!} = \frac{1}{k!}$.

**Recover $\mathcal{W}_0$ from $\mathcal{P}_0$.** $\forall (s, t) \in [c] \times [c]$, inverse 2D DFT is on $\mathcal{P}_0[:, :, s, t]$ to obtain $\mathcal{W}_0[:, :, s, t]$:

$$\mathcal{W}_0[:, :, s, t] = (\mathbf{F}_k^\top)^{-1} \mathcal{P}_0[:, :, s, t] \mathbf{F}_k^{-1} = \text{IDFT}_{2D}(\mathcal{P}_0[:, :, s, t]). \tag{16}$$

As $\mathcal{W}_0$ is the convolution kernel for the further convolution, to make the convolution compatible with mainstream real-value convolutional neural networks, we should keep $\mathcal{W}_0$ real. Based on the property of DFT, for inverse 2D discrete Fourier transform, the output $\mathbf{Y} \in \mathbb{R}^{k \times k}$ if purely real if and only if the input $\mathbf{X}$ is conjugate-symmetry, that is, $\mathbf{X}[i, j] = \mathbf{X}^*[(k-i)\%k, (k-j)\%k]$ for each $(i, j) \in [k] \times [k]$. Thus, to make $\mathcal{W}_0$ real, we need set $\mathcal{P}_0[i, j, :, :] := \mathcal{P}_0^*[(k-i)\%k, (k-j)\%k, :, :]$ for each $(i, j) \in [k] \times [k]$. Since $\mathcal{P}_0$ is real-value, we just need to set $\mathcal{P}_0[i, j, :, :]$ and $\mathcal{P}_0[(k-i)\%k, (k-j)\%k, :, :]$ identical. Due to the conjugate-symmetry settings, among these $k^2$ matrices $\{\mathcal{P}_0[(i, j, :, :]\}_{i=1, j=1}^{k, k}$, there are only $L$ different matrices. $L$ is $(k^2 + 1)/2$ when $k$ is odd and $(k^2 + 4)/2$ when $k$ is even. In the implementation, we encode all the information of $\mathcal{P}_0$ through a $L \times c \times c$ tensor $\hat{\mathcal{P}}_0$. Then we build $\mathcal{P}_0$ from $\hat{\mathcal{P}}_0$ by a mapping matrix $\mathbf{I} \in \mathbb{R}^{k \times k}$:

$$\mathcal{P}_0[i, j, :, :] = \hat{\mathcal{P}}_0[\mathbf{I}[i, j], :, :], \forall (i, j) \in [k] \times [k], \tag{17}$$

---

**Algorithm 1:** The proposed efficient orthogonal convolution. In training, the operations in lines 1-14 are involved in orthogonal convolution. In contrast, in evaluation, only the operation in line 14 is conducted, the operations in lines 1-13 have been conducted before evaluation and the obtained $\mathcal{W}$ has been stored in the model for further evaluation.

**Input:** Input tensor $\mathcal{X} \in \mathbb{R}^{n \times n \times c}$ where $c$ is the number of input channels, the spatial size of the convolution, $k$, the parameter tensor $\hat{\mathcal{P}}_0 \in \mathbb{R}^{L \times c \times c}$ (if $k$ is odd, $L$ is $\frac{k^2+1}{2}$ else $\frac{k^2+4}{2}$)

**Output:** Output tensor $\mathcal{Y} \in \mathbb{R}^{n \times n \times c}$ from conducting orthogonal convolution on $\mathcal{X}$.

1 **for** *all* $i \in [L]$ **do**
2      $\hat{\mathcal{P}}_0[i, :, :] \leftarrow \hat{\mathcal{P}}_0[i, :, :] - \hat{\mathcal{P}}_0[i, :, :]^\top$
3      $\hat{\mathcal{P}}_0[i, :, :] \leftarrow \text{SpectralNorm}(\hat{\mathcal{P}}_0[i, :, :])$
4      $\hat{\mathcal{P}}_0[i, :, :] \leftarrow \sum_{i=0}^{T} \frac{\hat{\mathcal{P}}_0[i, :, :]^i}{i!}$
5 **end**
6 $\mathbf{I} = \text{ConstructMap}(k)$ % construct the mapping matrix $\mathbf{I} \in \mathbb{R}^{k \times k}$
7 $\mathcal{P}_0 = \text{zeros}(k, k, c, c)$ ; $\mathcal{W}_0 = \text{zeros}(k, k, c, c)$
8 **for** *all* $(i, j) \in [k] \times [k]$ **do**
9      $\mathcal{P}_0[p, q, :, :] = \hat{\mathcal{P}}_0[\mathbf{I}[p, q], :, :]$
10 **end**
11 **for** *all* $(s, t) \in [c] \times [c]$ **do**
12      $\mathcal{W}_0[:, :, s, t] = \text{IDFT}_{2D}(\mathcal{P}_0[:, :, s, t])$
13 **end**
14 $\mathcal{Y} = \text{Conv}_{\mathcal{W}_0}(\mathcal{X}, \text{dilation} = n/k)$
15 **return** $\mathcal{Y}$

where $\mathbf{I}$ is a mapping matrix, which maps a pair $(i, j) \in [k] \times [k]$ to a scalar $l \in [1, L]$ and it satisfies

$$\mathbf{I}[i, j] = \mathbf{I}[(k-i)\%k, (k-j)\%k], \forall (i, j) \in [k] \times [k]. \tag{18}$$

The details of constructing the mapping matrix $\mathbf{I}$ is provided in Algorithm 2 of Appendix A.

**Dilation convolution.** In this step, we conduct the standard dilated convolution using the recovered kernel $\mathcal{W}_0$ obtained in Eq. (16) with a dilation $n/k$ on the input $\mathcal{X}$ and obtain the output $\mathcal{Y}$.

We summarize the proposed model in Algorithm 1. In each training iteration, given a batch of $B$ samples, the lines 1-13 are only conducted once, and the obtained $\mathcal{W}_0$ is shared across $B$ samples. After training, we store the computed $\mathcal{W}_0$ in line 12. In evaluation, we load the stored $\mathcal{W}_0$ and only compute the operation in line 14, which is just vanilla dilated convolution and thus is very efficient.

In the current configuration of Algorithm 1, we set $\mathcal{P}_0[p, q, :, :] \in \mathbb{R}^{c \times c}$ to be real matrix. In fact, $\mathcal{P}_0[p, q, :, :]$ can also be complex matrix. To make the output of the proposed orthogonal convolution compatible with existing mainstream real-value neural network, $\mathcal{W}_0[:, :, i, j] = \text{IDFT}_{2\text{D}}\mathcal{P}_0[:, :, i, j]$ should be a real matrix. That is, we should set $\mathcal{P}_0[:, :, i, j]$ to be conjugate symmetric and $\mathcal{P}_0[:, :, i, j]$ is unnecessarily real. When considering $\mathcal{P}_0[p, q, :, :]$ as a complex matrix, it makes the constructed orthogonal convolution kernel $\mathcal{W}_0$ more complete. Meanwhile, it brings more computational cost in training. In Appendix B, we compare with using a complex $\mathcal{P}_0[p, q, :, :]$.

## 5 IMPLEMENTATION

**Spectral normalization.** For a small approximation error from Taylor expansion, we conduct spectral normalization on $\mathbf{P}_0[p, q, :, :]$ to make $\|\mathbf{P}_0[p, q, :, :]\|_2 \leq 1$. Following Miyato et al. (2018); Singla & Feizi (2021a;b), we use the power method to estimate the approximated spectral norm and then divide $\mathbf{P}_0[p, q, :, :]$ by the estimated spectral norm.

**Padding.** Following SOC (Singla & Feizi, 2021b), we use cyclic padding to substitute zero-padding since zero-padded orthogonal convolutions must be size 1 (see proof in Appendix D.2 of SOC (Singla & Feizi, 2021b)). To be specific, for a dilated convolution of $k \times k$ size with a dilation $d$, we add $d(k-1)/2$ paddings on each side in a cyclic manner.

**Input and output channels.** The proposed orthogonal convolution is devised under the condition that the number of channels in input is equal to that in output, that is, $c_{\text{in}} = c_{\text{out}} = c$. When $c_{\text{in}} \neq c_{\text{out}}$, following SOC (Singla & Feizi, 2021b), we additionally conduct the following operations:

1. if $c_{\text{in}} > c_{\text{out}}$, we apply our orthogonal convolution with a kernel size of $k \times k \times c_{\text{in}} \times c_{\text{in}}$, generating the output with $c_{\text{in}}$ channels. We only select the first $c_{\text{out}}$ channels in the output.

2. if $c_{\text{in}} < c_{\text{out}}$, we use zero pad the input with $c_{\text{out}} - c_{\text{in}}$ channels. Then we apply our orthogonal convolution with a kernel size of $k \times k \times c_{\text{out}} \times c_{\text{out}}$.

**Strided convolution.** Given an input $\mathcal{X} \in \mathbb{R}^{n \times n \times c_{\text{in}}}$, a convolution with stride $s$ generates the output $\mathcal{Y} \in \mathbb{R}^{\frac{n}{s} \times \frac{n}{s} \times c_{\text{out}}}$. Our convolution is designed for 1-stride convolution. For a more general $s$-stride convolution, we decompose it into a down-sampling module and an orthogonal convolution, following BCOP (Li et al., 2019) and SOC (Singla & Feizi, 2021b). Specifically, the down-sampling module adapts the invertible down-sampling (Maduranga et al., 2019), which simply reshapes the input feature map $\mathcal{X}$ from $\mathbb{R}^{n \times n \times c_{\text{in}}}$ to $\mathbb{R}^{\frac{n}{s} \times \frac{n}{s} \times c_{\text{in}} s^2}$. The invertible down-sampling decreases the spatial size of the feature map from $k \times k$ to $k/s \times k/s$ and meanwhile strictly preserves the gradient norm. Then we conduct the proposed 1-stride orthogonal convolution on the reshaped $\mathcal{X}$.

**1-Lipschitz architecture.** Following SOC (Singla & Feizi, 2021b), we build a provably 1-Lipschitz network. The details of the architecture are given in Table 1. It consists of $L$ convolution layers in 5 blocks, and each block contains $L/5$ layers. By default, we only deploy the proposed ECO convolution in the last three blocks and keep the first two blocks as SOC. Each block consists of two groups of convolutions layers. The first group repeats $L/5 - 1$ convolutions layers with stride 1, and the second group contains a single convolution layer with stride 2. Following SOC, a MaxMin activation function (Anil et al., 2019) is added after each convolution layer. It has been implemented in the PaddlePaddle deep learning platform https://www.paddlepaddle.org.cn.

| | Input Size | Output Size | Group 1 | | Group 2 | |
|---|---|---|---|---|---|---|
| | | | Convolution | Repeats | Convolution | Repeats |
| **Block 1** | $24 \times 24$ | $12 \times 12$ | conv[3, 32, 1] | $(L/5 - 1)$ | conv[3, 64, 2] | 1 |
| **Block 2** | $12 \times 12$ | $6 \times 6$ | conv[3, 64, 1] | $(L/5 - 1)$ | conv[3, 128, 2] | 1 |
| **Block 3** | $6 \times 6$ | $3 \times 3$ | conv[3, 128, 1] | $(L/5 - 1)$ | conv[3, 256, 2] | 1 |
| **Block 4** | $3 \times 3$ | $1 \times 1$ | conv[3, 256, 1] | $(L/5 - 1)$ | conv[3, 512, 3] | 1 |
| **Block 5** | $1 \times 1$ | $1 \times 1$ | conv[1, 512, 1] | $(L/5 - 1)$ | conv[1, 1024, 1] | 1 |

Table 1: The configurations of the provably 1-Lipschitz architecture with $L$ convolution layers. It consists of 5 blocks. Each block contains two groups of convolution layers. The first group contains $(L/5 - 1)$ convolution layers of stride 1, and the second group contains a single convolution layer of stride 2. The output size decreases to $1/2$ after each block. conv $[k, c, s]$ denotes the convolution with the kernel of spatial size $k \times k$, output channels $c$, stride $s$. The dilation of a convolution layer is $n/k$ where $n$ is the spatial size of the input, and $k$ is the spatial size of the convolution kernel.

## 6 EXPERIMENTS

**Settings.** The training is conducted on a single NVIDIA V100 GPU with 32G memory. Following Singla & Feizi (2021b), the training takes 200 epochs. The initial learning rate is 0.1 when the number of convolution layers is larger than 25 and 0.05 otherwise, and it decreases by a factor of 0.1 at the 50th and the 150th epoch. We set the weight decay as 5e-4.

**Evaluation metrics.** We evaluate the performance proposed orthogonal convolution by two metrics: standard recognition accuracy and the robust recognition accuracy. The standard recognition accuracy is the same as that used in classic image classification. In contrast, the robust recognition accuracy is the accuracy under an $l_2$ perturbation. As shown in Section 3.3, $L$-Lipschitz network has the robustness certificate when the $\ell_2$-norm perturbation $\epsilon$ is less than $\mathcal{M}_n(\mathbf{x})/\sqrt{2}L$. Thus, for a 1-Lipschitz we built based on the proposed ECO convolution has the robustness certificate as

| Model | Conv. Type | CIFAR10 | | CIFAR100 | | Time per Epoch (s) |
|---|---|---|---|---|---|---|
| | | Standard Accuracy | Robust Accuracy | Standard Accuracy | Robust Accuracy | |
| LipConvnet-5 | BCOP | 70.48% | 51.11% | 39.19% | 24.54% | 49.6 |
| | SOC | 71.31% | 52.11% | 38.42% | 23.68% | 18.1 |
| | Ours | 71.69% | 52.36% | 38.68% | 23.86% | 18.8 |
| LipConvnet-10 | BCOP | 70.79% | 52.68% | 39.29% | 24.99% | 73.0 |
| | SOC | 72.12% | 54.72% | 39.13% | 25.01% | 27.9 |
| | Ours | 72.15% | 54.57% | 39.64% | 24.83% | 29.3 |
| LipConvnet-15 | BCOP | 70.48% | 51.87% | 37.93% | 23.94% | 86.2 |
| | SOC | 72.30% | 55.34% | 39.26% | 24.76% | 33.9 |
| | Ours | 72.87% | 55.20% | 39.43% | 24.74% | 40.3 |
| LipConvnet-20 | BCOP | 69.99% | 51.96% | 36.37% | 22.90% | 116.8 |
| | SOC | 72.23% | 55.45% | 39.24% | 25.29% | 39.2 |
| | Ours | 72.52% | 55.41% | 40.03% | 25.36% | 44.9 |
| LipConvnet-25 | BCOP | 68.84% | 50.41% | 33.04% | 19.72% | 138.5 |
| | SOC | 72.30% | 55.34% | 38.77% | 24.05% | 46.3 |
| | Ours | 72.42% | 55.32% | 38.32% | 24.04% | 49.2 |
| LipConvnet-30 | BCOP | 69.77% | 51.26% | 30.95% | 18.56% | 150.2 |
| | SOC | 72.80% | 55.71% | 39.55% | 25.42% | 54.9 |
| | Ours | 72.48% | 55.46% | 39.18% | 24.41% | 57.7 |
| LipConvnet-35 | BCOP | 68.61% | 49.56% | 27.75% | 13.73% | 166.6 |
| | SOC | 72.89% | 55.71% | 38.12% | 24.32% | 60.6 |
| | Ours | 71.59% | 54.76% | 38.60% | 24.14% | 64.0 |
| LipConvnet-40 | BCOP | 67.64% | 48.49% | 27.09% | 13.71% | 180.3 |
| | SOC | 72.10% | 55.33% | 37.98% | 23.63% | 67.3 |
| | Ours | 72.43% | 56.08% | 38.24% | 24.35% | 70.9 |

Table 2: Comparisons among BCOP (Li et al., 2019), SOC (Singla & Feizi, 2021b) and our ECO convolution on standard accuracy, robust accuracy, the time cost per epoch in training.

$\mathcal{M}_n(\mathbf{x})/\sqrt{2} \geq \epsilon$. Following Li et al. (2019) and Singla & Feizi (2021b), when testing the robust accuracy, we set the default norm of perturbation, $\epsilon$, as $36/255$. In this case, the robust accuracy is the ratio of samples satisfying $\mathcal{M}_n(\mathbf{x})/\sqrt{2} \geq 36/255$.

**Standard and robust recognition accuracy.** Table 2 compares the recognition accuracy of the proposed ECO convolution with BCOP (Li et al., 2019) and SOC (Singla & Feizi, 2021b) on CIFAR10 and CIFAR100 datasets. As we can observe from the table that our ECO convolution achieves competitive performance compared with BCOP and SOC in both standard and robust recognition accuracy. Specifically, on CIFAR-10 and CIFAR-100, when the number of convolution layers, $L$, is 5 or 40, ours outperforms both SOC and BCOP in standard and robust recognition accuracy.

**Ablation study on the dilated convolution.** The dilation of the proposed ECO convolution is $n/k$ where $n$ is the spatial size of the feature map and $k$ is the spatial size of the convolution kernel. In an early convolution block, the feature map is of large size, and thus the dilation of the convolution is large. We discover that the large dilation in early blocks deteriorate the recognition performance. We conduct an ablation study on the dilated convolution by fixing the first $l$ blocks as SOC and only use our SOC in the last $5-l$ blocks in LipConvnet-5. As shown in Table 3, when we deploy our ECO convolution is the first one or two convolution blocks, the performance becomes worse considerably. Thus, we only apply our ECO convolution in the last three convolution blocks, by default.

| $l$ | 0 | 1 | 2 | 3 | 4 | 5 |
|---|---|---|---|---|---|---|
| CIFAR10 Standard | 67.88% | 70.97% | 71.69% | 71.64% | 71.54% | 71.31% |
| CIFAR10 Robust | 48.83% | 51.66% | 52.36% | 52.55% | 52.08% | 52.11% |
| CIFAR100 Standard | 31.35% | 38.34% | 38.68% | 38.81% | 38.82% | 38.42% |
| CIFAR100 Robust | 21.60% | 23.40% | 23.86% | 24.02% | 38.85% | 23.68% |

Table 3: Ablation on implementing the first $l$ blocks by SOC and the last $5-l$ blocks by our ECO convolution. We use LipConvnet-5 model.

**Training and evaluation time.** As shown in the last column of Table 2, SOC is slightly more efficient than our ECO convolution in training. It is because the dilation convolution in our ECO adds more padding, making the input feature map larger than that in SOC. Meanwhile, as shown in Table 4, our ECO convolution is faster than SOC in evaluation. It is due to that we have already constructed the ECO convolution before evaluation. Thus, in evaluation, the computational cost of our ECO convolution is the same as standard convolution. In contrast, SOC needs to take $T$ times convolution operations per convolution layer, taking $T$ times computational cost as standard convolution. By default, SOC sets $T = 5$ in training and $T = 10$ in evaluation.

| Model | LC-5 | LC-10 | LC-15 | LC-20 | LC-25 | LC-30 | LC-35 | LC-40 |
|---|---|---|---|---|---|---|---|---|
| SOC | 0.162s | 0.276s | 0.373s | 0.480s | 0.587s | 0.693s | 0.774s | 0.890s |
| **ECO** | **0.106**s | **0.108**s | **0.116**s | **0.124**s | **0.130**s | **0.135**s | **0.142**s | **0.147**s |

Table 4: Comparisons between SOC (Singla & Feizi, 2021b) and our ECO convolution on the evaluation time on the third convolution block. LC denotes LipConvnet. In evaluation, we set the batch size as 128 and the evaluation is conducted on a single V100 GPU.

## 7 CONCLUSION

This work investigates building the convolution layer with an orthogonal Jacobian matrix, which is 1-Lipschitz in the 2-norm and preserves the gradient. By exploiting the relation between the convolution kernel and the Jacobian matrix, we explicitly construct the convolution kernel to achieve the orthogonal convolution. The explicitly constructed orthogonal convolution (ECO) takes the exact computational cost as a standard dilated convolution in evaluation, which is more efficient than the existing state-of-the-art method, SOC, approximating the orthogonal convolution through multiple times convolution operations. Our experiments on CIFAR-10 and CIFAR-100 show that our ECO achieves competitive standard and robust accuracy as SOC using much less evaluation time.

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

## A    CONSTRUCTING THE MAPPING MATRIX **I**

In Algorithm 2, we give the details of constructing the mapping matrix **I**.

---

**Algorithm 2:** Constructing the mapping matrix **I**

---
**Input:** The spatial size of the convolution kernel, $k$.
**Output:** The mapping matrix **I**
1  $\mathbf{I} = \text{zeros}(k, k)$;
2  count = 0;
3  **for** *all* $(i, j) \in [k] \times [k]$ **do**
4    **if** $\mathbf{I}[(k-i)\%k, (k-j)\%k] == 0$ **then**
5      $\mathbf{I}[i, j] = \text{count}$;
6      count = count + 1;
7    **end**
8    **else**
9      $\mathbf{I}[i, j] = \mathbf{I}[(k-i)\%k, (k-j)\%k]$;
10   **end**
11 **end**
12 **return I**

---

## B    COMPARISONS WITH THE COMPLEX SETTINGS

As we mentioned in Section 4, to obtain a real-value convolution kernel $\mathcal{W}_0$, the entries in $\mathcal{P}_0$ are unnecessarily real. We compare the performance when $\mathcal{P}_0$ is constrained to be real as Algorithm 1 and that when $\mathcal{P}_0$ is a complex tensor. As shown in Table 5, on CIFAR-10 benchmark, using complex $\mathcal{P}_0$, generally, it performs slightly better than its real counterpart on both standard accuracy and robust accuracy. The better performance is attributed to the fact that the complex setting can construct more complete orthogonal convolution. It is worth noting that the computational cost of multiplication between complex matrices is four times as that between real matrices of the same sizes. But the complex $\mathcal{P}_0$ is only used for constructing the real orthogonal kernel $\mathcal{W}_0$. After kernel construction, the computation cost of convolution is the same for the layer with complex $\mathcal{P}_0$ and that with real $\mathcal{P}_0$. Thus, as shown in Table 5, the training time per epoch when using complex $\mathcal{P}_0$ is around 2 times as that using real $\mathcal{P}_0$. In evaluation, since we have constructed the real $\mathcal{W}_0$ from a real or complex $\mathcal{P}_0$ before evaluation, the computational cost is the same no matter $\mathcal{P}_0$ is complex and real.

|  | Standard Accuracy | | Robust Accuracy | | Training Time | |
|---|---|---|---|---|---|---|
| **Model** | **LC-10** | **LC-15** | **LC-10** | **LC-15** | **LC-10** | **LC-15** |
| **Real** | 72.15% | 72.87% | 54.57% | 55.20% | 40.3 | 44.9 |
| **Complex** | 72.40% | 72.98% | 54.60% | 55.53% | 78.0 | 81.8 |

Table 5: Comparisons between the real and complex $\mathcal{P}_0$ on CIFAR-10 dataset. LC is the abbreviation for LipConvnet.

## C    COMPARISONS WITH THE STANDARD CONVOLUTION

Here, we compare the proposed orthogonal convolution with the standard convolution without orthogonal constraint. To make a fair comparison, we only replace the orthogonal convolution layers of LipConvnet-5 by the standard convolution and keep the other layers unfixed. Since the network built upon the standard convolution cannot preserve the 1-Lipschitz property, we cannot directly obtain the robust accuracy based on $\mathcal{M}_n(\mathbf{x})$ as the original LipConvnet-5 network. Thus, we utilize Projected Gradient Descent (PGD) attack (Madry et al., 2018) to obtain the robust accuracy upper

|  | CIFAR10 | | CIFAR100 | |
|---|---|---|---|---|
|  | Standard Accuracy | Robust Accuracy | Standard Accuracy | Robust Accuracy |
| Standard | **75.21**% | 9.63% | **45.62**% | 7.31% |
| ECO | 71.69% | **52.36**% | 38.42% | **23.68**% |

Table 6: Comparisons with the standard convolution.

bound under the distortion with the $\ell_2$ norm less than $\frac{36}{255}$. The results are shown in Table 6. As shown in the table, on both CIFAR10 and CIFAR10 datasets, the network based on the standard convolution achieves higher standard accuracy than the network based on our ECO convolution. In the meanwhile, ours achieves higher robust accuracy on both datasets under the same scale of distortion.

## D  TRUNCATED TAYLOR EXPANSION

Note that, the exact matrix exponential (Casado & Martínez-Rubio, 2019) $\exp(\mathbf{A})$ relies on singular value decomposition (SVD), which is not friendly for GPU computing platform. Thus, we adopt the truncated Taylor Expansion (Singla & Feizi, 2021b), $\mathbf{S}_T(\mathbf{A}) = \sum_{i=0}^{T} \frac{\mathbf{A}^i}{i!}$, to approximate the exact matrix exponential $\exp(\mathbf{A})$. The truncation error $\mathbf{R} = \exp(\mathbf{A}) - \mathbf{S}_T(\mathbf{A})$ satisfies

$$\|\mathbf{R}\|_2 = \|\exp(\mathbf{A}) - \mathbf{S}_T(\mathbf{A})\|_2 \leq \frac{\|\mathbf{A}\|_2^T}{T!}. \tag{19}$$

Below we first compare the time cost in GPU platform using the exact matrix exponential $\exp(\mathbf{A})$ and that based on the truncated Taylor Expansion $\mathbf{S}_T(\mathbf{A})$. To be specific, we implement $\exp(\mathbf{A})$ simply based on the torch.matrix_exp. We measure the time cost on GPU using the function torch.cuda.Event and we set $T = 10$ in truncated Taylor Expansion $\mathbf{S}_T(\mathbf{A})$. As shown in Table 7, the latency based on the truncated Taylor Expansion is significantly less than that based on the exact matrix exponential.

| Exact | Truncated |
|---|---|
| 12.8ms | 2.5ms |

Table 7: The latency of the exact matrix exponential and the truncated Taylor Expansion in a layer.

Then we show the scale of the average factual truncated error $\mathbf{R}$ with respect to $T$. It is based on the trained LipConvnet-5 network built upon the proposed ECO convolution. By default, we set $T = 5$ in training and $T = 10$ in testing. As shown in Table 8, the scale of the truncation error is relatively small when $T$ is 5 or 10. Meanwhile, we also show the influence of the truncation error on the orthogonality of the convolution. Following Trockman & Kolter (2021), we measure the orthogonality of a convolution layer through $\frac{\|\text{vec}(\text{conv}(\boldsymbol{\mathcal{X}}))\|_2}{\|\text{vec}(\boldsymbol{\mathcal{X}})\|_2} - 1$, and a strict orthogonal convolution layer should satisfy

$$\frac{\|\text{vec}(\text{conv}(\boldsymbol{\mathcal{X}}))\|_2}{\|\text{vec}(\boldsymbol{\mathcal{X}})\|_2} - 1 = 0. \tag{20}$$

We estimate the deviation of the proposed ECO convolution from the strict orthogonal convolution through this measurement. As shown in Table 8, when $T$ is 5 or 10, the deviation of our ECO convolution from the strict orthogonal convolution is relative marginal.

| $T$ | 1 | 2 | 3 | 4 | 5 | 10 |
|---|---|---|---|---|---|---|
| $\|\mathbf{R}\|_2$ | 0.48 | 0.16 | 0.04 | 0.008 | 0.001 | 0.0001 |
| $\frac{\|\text{vec}(\text{conv}(\boldsymbol{\mathcal{X}}))\|_2}{\|\text{vec}(\boldsymbol{\mathcal{X}})\|_2} - 1$ | 0.12 | 0.019 | 0.0038 | 0.0004 | 0.0000 | 0.0000 |

Table 8: The scale of the average factual truncated error.

