# OpenReview forum: "Constructing Orthogonal Convolutions in an Explicit Manner"
_ICLR.cc/2022/Conference — ICLR 2022 Poster_

### Official Review · Reviewer_YtM2 · 2021-11-03

**Correctness:** 4
**Technical Novelty And Significance:** 4
**Empirical Novelty And Significance:** 4
**Recommendation:** 8
**Confidence:** 4

**Main Review:**

Technical comments:

- It is not clear to me why one needs to consider the *expanded* convolution kernel. In principle, the orthogonality of a convolutional kernel is independent of the size of the feature map that it is applied to. However, the construction of orthogonal convolution in the paper requires one to first expand the kernel to the size of the feature map that it is applied to, by padding zeros. My understanding that this expansion causes a lot of trouble that requires some intricate fixes, such as Eq. (12), (13), and (14), that greatly complicates the method.

- Can the authors comment on the completeness of their parameterization for orthogonal convolution? Namely, does the parameterization contain the set of all orthogonal convolution, or just a subset?

- In Eq. (3), it seems that $F_k$ should be $F_n$ since $\mathcal{W}$ is of size $n \times n$.

Experiments:

- Table 2: It could be good to add a baseline where no orthogonality is enforced.

- Can the authors comment on why ECO produces significantly better performance than SOC, in Table 2? If both of them enforce strict orthogonality of convolution then shouldn't they be giving exactly the same accuracy (in principle)? BCOP is inferior probably because it only parameterizes a subset of all orthogonal convolution (or maybe the authors have other explanations?)

- The caption of Table 3 says that it reports evaluation time, while the text says that it is the training time. Am I missing anything here?

- The experiments only contain results with LipConvnet which are perhaps non-standard and have far worse performance than standard networks, such as ResNet. This makes it less convincing as to how important the method is for practice. Are there any results on standard networks?

Clarity:

- "simply constructing an orthogonal convolution kernel cannot achieve an orthogonal convolution": Not sure what this mean. How is "orthogonal convolution kernel" different from "orthogonal convolution"? Actually, what do those two terms mean, exactly?

Other comments:

- While the paper argues that previous methods for enforcing orthogonality in convolutions, such as Trockman & Kolter '21 and Singla & Feizi '21, have computational issues, it fails to mention that there are earlier work [a] as well as more recent work [b] that are computationally more efficient.

[a] Deep Isometric Learning for Visual Recognition, ICML 2020
[b] Scaling-up Diverse Orthogonal Convolutional Networks with a Paraunitary Framework, 2021

**Summary Of The Paper:**

This paper presents a method for enforcing strict orthogonality for convolutional layers. The method is based on considering the spectral domain, where the orthogonality of the 4D conv kernel (in spatial domain) is characterized as the orthogonality of 2D matrices, for which orthogonality can be enforced by existing techniques such as Carley transform. It is shown in experiments that this method is significantly faster than the recent skew orthogonal convolution (at ICML'21) method.

**Summary Of The Review:**

This is a nicely written paper for enforcing orthogonality of convolutional kernels with a computationally efficient method. However, I find the experiments to be shallow relative to related works, and cannot fully justify the practical value of the proposed method. I also have some minor concerns on the technical approach, that hopefully can be clarified during rebuttal.

---

> ### Author Response · Authors · 2021-11-12
> **Initial response to Reviewer YtM2**
>
> We appreciate the valuable suggestions from the reviewer, which are really helpful to enhance the quality of the submitted manuscript. To receive more constructive suggestions from you in time, we first clarify some issues you raise in this initial response. At the same time, we are revising the manuscript. Please look forward to our revised manuscript in the next few days. We also look forward to your reply after your read our initial response.
>
> **Q1: Why  consider the expanded convolution kernel? The orthogonality of a convolutional kernel is independent of the size of the feature map. However, the construction of orthogonal convolution in the paper requires one to first expand the kernel to the size of the feature map that it is applied to, by padding zeros. My understanding is that this expansion causes a lot of trouble that requires some intricate fixes, such as Eq. (12), (13), and (14), which greatly complicates the method.**
>
> A1: Thanks for your question, which involves the core part of the proposed method.  First of all, we clarify that, in this work, we seek to achieve an orthogonal Jacobian matrix instead of an orthogonal convolutional kernel.  Secondly, the orthogonality of a Jacobian matrix is dependent on the size of the feature map. In fact, the Jacobian matrix itself is dependent on the size of the feature map. It might be easier to get this point by considering the dimension of a Jacobian matrix. Let us denote the dimension of the input feature map by $n_{in} \times n_{in} \times c_{in}$ and the dimension of the output feature map by $n_{out} \times n_{out} \times c_{out}$. In this case,  the dimension of the Jacobian matrix is $n_{in}^2c_{in} \times n_{out}^2c_{out}$.  For the input feature maps of different scales, the dimensions of the Jacobian matrix will be different.  Thirdly, let us explain why we need to expand the kernel to the size of the feature map that it is applied to, by padding zeros. Our method is built upon the Theorem 6 in [1] (Hanie Sedghi et al. 2019). We follow the operation in the first paragraph of section 2.1 in [1] (Hanie Sedghi et al. 2019) to embed this $k \times k$ matrix in an $n \times n$ matrix, by padding with zeroes (which corresponds to the fact that the offsets with those indices are not used).  If we do not pad zeros,  Theorem 6 in [1] will not hold.
>
> [1] The Singular Values of Convolutional Layers
>
>
>
> **Q2: The completeness for orthogonal convolution. Does it contain the set of all orthogonal convolution, or just a subset?**
>
> A2:  Our method only covers a subset of the set of all orthogonal convolutions. By default, we constrain $\mathcal{P}_0[p,q,:,:]$  to be a real-value matrix. In fact, $\mathcal{P}_0[p,q,:,:]$  is not necessarily real.  As we mentioned in the paragraph above Table 4 of the originally submitted manuscript, $\mathcal{P}_0[p,q,:,:]$ can also be complex matrix.   When considering $\mathcal{P}_0[p,q,:,:]$ as a complex matrix, it makes the constructed orthogonal convolution kernel ${\mathcal{W}}_0$ more complete. Meanwhile, it brings more computational cost in training. In Appendix B, we compare with using a complex $\mathcal{P}_0[p,q,:,:]$.
>
>
> **Q3: In Eq. (3), $\mathbf{F}_k$ should be $\mathbf{F}_n$  since $\mathbf{W}$ is of size $n\times n$.**
>
> A3: Yes, $\mathbf{F}_k$ should be $\mathbf{F}_n$ . We will fix it in the revised manuscript.
>
> **Q4: Table 2: It could be good to add a baseline where no orthogonality is enforced.**
>
> A4: We will add a baseline where no orthogonality is enfored in the revised manuscript. Please look forward to our revised manuscript in a few days.
>
> **Q5:  Why ECO produces better performance than SOC in Table 2? If both of them enforce strict orthogonality of convolution then shouldn't they be giving exactly the same accuracy (in principle)?**
>
> A5: Our orthogonal convolution is not equivalent to that in ECO. The key difference is that our convolution is a dilated convolution.
>
> **Q6: The caption of Table 3 says that it reports evaluation time, while the text says that it is the training time. Why?**
>
> A6: The caption of Table 3 is correct, and the text has typos. We will correct the texts in the revised manuscript.
>
>
> **Q7: The experiments only contain results of LipConvnet, which are perhaps non-standard and have far worse performance than standard networks, such as ResNet. This makes it less convincing as to how important the method is for practice. Any results on standard networks?**
>
>
> A7: The residual connection in ResNet is not 1- Lipschitz. Therefore, even though we built orthogonal convolution to make the convolution 1- Lipschitz,  the network which combines the orthogonal convolution with the residual connection is not 1-Lipschitz, and cannot achieve the certified robust as the LipConvnet. But we will do the experiments by replacing the standard convolution with the orthogonal convolution in ResNet and update them in the revised manuscript.  Please look forward to our revised manuscript in a few days.

---

> ### Author Response · Authors · 2021-11-18
> **The second response to Reviewer YtM2**
>
> Dear Reviewer, we thank you again for your valuable suggestions to help improve the manuscript. Below we give more detailed clarification and look forward to more feedback from you.
>
> **Q8: "simply constructing an orthogonal convolution kernel cannot achieve an orthogonal convolution": What this means. How is "orthogonal convolution kernel" different from "orthogonal convolution"?
>
> A8:  An **orthogonal convolution kernel** is a convolution kernel that can be expanded into an orthogonal matrix. To be specific, let us denote a convolution kernel by ${\mathcal{W}} \in \mathbb{R}^{k \times k \times c_{\mathrm{in}} \times  c_{\mathrm{out}} }$ where $k$ is the size of local reception field, $c_{\mathrm{in}} $ is the channel number of the input feature map, and  $c_{\mathrm{out}} $ is the channel number of the input feature map. We can expand the higher-order tensor $\mathcal{W}$ into a two-order matrix $\mathbf{W} \in \mathbb
> {R}^{k^2c_{\mathrm{in}}  \times c_{\mathrm{out}}}$. Given a  $k\times k \times c_{\mathrm{in}}$  patch,  $\mathcal{P}$,  the convolution generates an output vector:
>
> \begin{equation}
> \mathbf{y} = \mathrm{vec}(\mathcal{P}) \mathbf{W}.
> \end{equation}
>
>
> In this case,  the local Jacobian matrix from  $\mathbf{y}$ to  $\mathrm{vec}(\mathcal{P})$ is computed by
> \begin{equation}
>  \mathbf{J}_{local} = \frac{\partial \mathbf{y}}{\partial \mathrm{vec}(\mathcal{P})} = \mathbf{W}.
> \end{equation}
> Thus, we only need to make $\mathbf{W}$  orthogonal to achieve an orthogonal local Jacobian matrix.
> But the orthogonality of the local Jacobian matrix is not equivalent to the orthogonality of the global Jacobian matrix.
>
>
>  We define an **orthogonal convolution** as the convolution with an orthogonal global Jacobian matrix $\mathbf{J}_{global}$. To be specific, let us denote the whole input feature map by $\mathcal{X}$ and the whole output feature map by  $\mathcal{Y} = convolution(\mathcal{X})$,   the global Jacobian matrix is
>
> \begin{equation}
> \mathbf{J}_{{global}}= \frac{\partial  \mathrm{vec}(\mathcal{Y})}{\partial \mathrm{vec}(\mathcal{X})}.
> \end{equation}
> In a nutshell, the **orthogonal convolution** with an orthogonal global Jacobian matrix is different from **orthogonal convolution kernel** with an orthogonal local Jacobian matrix.
>
> **Q9: While the paper argues that previous methods for enforcing orthogonality in convolutions, such as Trockman & Kolter '21 and Singla & Feizi '21, have computational issues, it fails to mention that there is earlier work [a] as well as more recent work [b] that are computationally more efficient.**
>
> [a] Deep Isometric Learning for Visual Recognition, ICML 2020 [b] Scaling-up Diverse Orthogonal Convolutional Networks with a Paraunitary Framework, 2021
>
>
> A9: In the revised manuscript, we have discussed the relations between the earlier work [a] and [b] in Section 2 (related work). To be specific, the [a] encourages the orthogonality by a regularizer, which is efficient but cannot achieve the strict orthogonality and the provable robustness.
> [b] is indeed good work that achieves orthogonal convolutions via paraunitary systems in a very elegant manner. However, as shown in Figure 4 (a) in [b], it is slower than BCOP.  In contrast, ours is faster than BCOP as shown in Table 2 of our submitted manuscript.
>
>
> **Q10:   It could be good to add a baseline where no orthogonality is enforced.**
>
> A10: We have added the results of the baseline where no orthogonality is enforced in Table 6 of Appendix C.
>
> **Q11:   Why consider the expanded convolution kernel? In principle, the orthogonality of a convolutional kernel is independent of the size of the feature map that it is applied to. However, the orthogonal convolution in the paper requires expanding the kernel to the size of the feature map by padding zeros. My understanding that this expansion causes a lot of trouble that requires some intricate fixes, such as Eq. (12), (13), and (14), that greatly complicates the method**
>
> A11: We want to highlight again, it is necessary to expand the convolution kernel from $k\times k$ to $n\times n$ before we apply DFT in Theorem 1 of our manuscript and  Theorem 6 in [c] (Hanie Sedghi et al. ICLR 2019). This is also consistent with [d]  (Asher Trockman et al. ICLR’21) which defines $n\times n$ convolution kernel instead of $k\times k$ convolution kernel, where $n$ is the spatial size of the input feature map.  We agree with you that the expanded convolution kernel makes the problem more challenging. Besides, we highlight again that the orthogonality of a convolutional kernel is indeed dependent on the size of the feature map.
>
> [c] The Singular Values of Convolutional Layers, ICLR'19
> [d] Orthogonalizing Convolutional Layers with the Cayley Transform, ICLR'21
>
> **Q12:  Experiments only on LipConvnet make it less convincing for how important it is for practice.  Any results on standard networks such as ResNet?**
>
> A12:   In Appendix D, we add Table 7 to show the performance on the ResNet-18.

---

> > ### Comment · Reviewer_YtM2 · 2021-11-21
> > **"Orthogonality of a convolutional kernel is indeed dependent on the size of the feature map."**
> >
> > Thanks for the detailed response. All of my questions/concerns have been adequately addressed except for the concern on the necessity of expanding the convolutional kernels to the shape of the input feature map. In particular the authors argued that
> >
> > >Orthogonality of a convolutional kernel is indeed dependent on the size of the feature map
> >
> > I do not think this statement is correct. [a, b, c] all studied orthogonal convolution and none of their methods required the size of the feature map to enforce orthogonality. Part of the reason that the authors make this argument seems that their derivation is based on first expressing the convolution operator in the form of a (2D block-circulant) matrix then enforcing orthogonality of the matrix, and that the shape of this matrix is dependent on the size of the feature map. However, note that there are many zero regions in such a matrix and a careful analysis (see e.g. [a]) will show that orthogonality does not depend on the shape of the feature map. Alternatively, one can also derive orthogonality of convolution from definition directly [b] or by going to the frequency domain [c], where there is no need of expanding 4D kernels into 2D block circulant matrices hence easier to see that the notion of orthogonality does not depend on feature map shape.
> >
> > [a] Orthogonal convolutional neural networks, CVPR 19
> > [b] Deep Isometric Learning for Visual Recognition, ICML 2020
> > [c] Scaling-up Diverse Orthogonal Convolutional Networks with a Paraunitary Framework, 2021

---

> > > ### Author Response · Authors · 2021-11-21
> > > **Response to "Orthogonality of a convolutional kernel is indeed dependent on the size of the feature map."**
> > >
> > > Dear Reviewer YtM2,
> > >
> > > We thank you for your encouraging feedbacks that most of your questions have been addressed. Below let us focus on the only question to the argument that "Orthogonality of a convolutional kernel is indeed dependent on the size of the feature map".
> > >
> > > We carefully read the mentioned papers [a,b] again and agree with you that in [a,b], the orthogonality is independent of the feature map size. However, we want to clarify that the setting in [a,b] is slightly different from ours. To be specific, in [a,b], given an input feature map of shape $n\times n$ and a convolution kernel of shape $k\times k$ with stride 1, it generates the output feature map of the shape $(n-k-1)\times (n-k-1)$ (as visualized in Figure 3 in [a]). It is worth noting that it does not additionally add paddings in the input feature map to make the shape of the output feature map equal to the shape of the input feature map.  That is,  the Jacobian matrix generated from [a] (when $k>$1) is no longer  a square matrix and does not achieve an orthogonal Jacobian matrix (an orthogonal matrix $\mathbf{J}$ must be a square matrix  https://en.wikipedia.org/wiki/Orthogonal_matrix. A non-square matrix can never satisfy $\mathbf{J}\mathbf{J}^{\top} = \mathbf{J}^{\top}\mathbf{J}= \mathbf{I}$ and it might achieve $\mathbf{J}\mathbf{J}^{\top} =\mathbf{I}$ or  $\mathbf{J}^{\top}\mathbf{J}= \mathbf{I}$ at most).  Thus, it only enforces the row orthogonality ( $\mathbf{J}\mathbf{J}^{\top} =\mathbf{I}$) or the column orthogonality ($\mathbf{J}^{\top}\mathbf{J}= \mathbf{I}$) for the Jacobian matrix, as illustrated in Section 3.2 of [a]. The row orthogonality or column orthogonality achieved in [a]  are not equivalent to the matrix orthogonality ($\mathbf{J}\mathbf{J}^{\top} = \mathbf{J}^{\top}\mathbf{J}= \mathbf{I}$).
> > >
> > >
> > > In contrast, in [d], [e], [f], and our ECO convolution, to make the shape of the output feature map equal to that of the input feature map,  additional  $k-1$ row and column paddings are added in a cyclic manner, leading to a Jacobian matrix in **doubly block circulant** structure, whereas the Jacobian matrix in [a,b] is in **doubly block Toeplitz** structure.  In [d], [e], [f], and our ECO convolution, we achieve the strict matrix orthogonality in Jacobian matrix ($\mathbf{J}\mathbf{J}^{\top} = \mathbf{J}^{\top}\mathbf{J}= \mathbf{I}$).  It is worth noting that, the padding in [d], [e], [f], and our ECO convolution  must be cyclic padding instead of zero padding since the zero padding can only support the orthogonal convolution of shape $1\times 1$ as proved in Appendix D.2 of [d]. When taking the cyclic padding into consideration, the condition in Eq. (2) of [a] is no longer equivalent to the condition in Eq. (3) of [a], and the orthogonality is no longer independent to the size of the input feature map.  Our analysis on [a] is also applied to [b] since these two works adopt a similar manner for achieving orthogonality/isometry.  In the meanwhile, [c] also ignores the paddings.
> > >
> > > In summary, when the spatial shape of the convolution kernel  $k>1$,  if we do not use cyclic paddings and we only seek to achieve row/column orthogonality  (the Jacobian matrix is not a square matrix) as [a,b], the row/column  orthogonality of the Jacobian matrix is independent to the size of the input feature map.  In contrast, if we use cyclic paddings  to achieve the strict matrix orthogonality (the Jacobian matrix is a square matrix), the orthogonality of the Jacobian matrix is dependent on the size of the input feature map.
> > >
> > >
> > > We believe the raised question by the reviewer is very important and very worthy of being discussed, and we are also further investigating this point to ensure our claim is correct. We also look forward to your feedback when you read our response to this point. Please kindly correct us if we misunderstand some points.
> > >
> > >
> > > [a] Orthogonal convolutional neural networks, CVPR 2019
> > >
> > > [b] Deep Isometric Learning for Visual Recognition, ICML 2020
> > >
> > > [c] Scaling-up Diverse Orthogonal Convolutional Networks with a Paraunitary Framework, 2021
> > >
> > > [d] Preventing Gradient Attenuation in Lipschitz Constrained Convolutional Networks, NIPS 2019
> > >
> > > [e] Orthogonalizing Convolutional Layers with the Cayley Transform, ICLR 2021
> > >
> > > [f] The Singular Values of Convolutional Layers, ICLR 2019

---

> > > > ### Author Response · Authors · 2021-11-22
> > > > **Additional Response to Response from Reviewer YtM2**
> > > >
> > > > Hi, Dear Reviewer YtM2
> > > >
> > > > Thanks again for your great efforts and insightful comments for improving the submitted manuscript.
> > > > Since the discussion deadline is approaching, we wonder if our response is clear? Do we need more clarifications?
> > > >
> > > > Best,
> > > >
> > > > Anonymous Authors

---

> > > > ### Comment · Reviewer_YtM2 · 2021-11-27
> > > > **Concern on the role of kernel size is clarified [Updated]**
> > > >
> > > > I would like to thank the authors for the detailed response regarding the role of kernel size on the notion of orthogonality. I read more into the references and here are my understanding:
> > > >
> > > > 1) I agree with the authors' comments on [a] that it does not use padding and hence only enforces row or column orthogonality.
> > > >
> > > > 2) I do not agree with the comment that [b] and [a] adopt a similar manner for achieving orthogonality/isometry. Specifically, [b] explicitly mentions that it uses zero padding on the feature map (Sec. 2.2, Notations), hence it does not have the issue in [a]. Therefore, [b] derives the notion of orthogonality for convolution (Thm 1) that is independent of the feature map.
> > > >
> > > > 3) One may argue that this conclusion in [b] cannot possibly hold true since [d] says that orthogonality cannot be achieved with zero padding on the feature map (unless kernel size = 1). My conjecture here is that although Thm 1 in [b] derives equivalent conditions for orthogonality, those conditions cannot hold true unless kernel size = 1.
> > > >
> > > > 4) However, I believe this issue of [b] can be easily fixed: Although [b] says it uses zero padding of feature map, the same derivation and conclusion will follow if zero padding is replaced by cyclic padding. After applying this fix, Thm 1 in [b] provides the notion of orthogonality for (cyclic) convolution that is independent of the feature map.
> > > >
> > > > 5) Moving to [c], I do not agree with the statement that it ignores padding. In the proof to Theorem A.5 it explicitly states that they do circular padding for finite inputs. Hence, [c] is another example where the notion of orthogonality is independent of kernel size.
> > > >
> > > > These being said, there are also many works such as [d, e, f] where zero padding is needed for deriving their methods. Hence, I would say there is nothing technically wrong with the proposed method (though it may cause extra troubles in the derivation as I commented in the original review), so should not be a reason to reject the paper. I hence raise my score. But I do suggest the authors to be careful in claiming that padding the kernel is necessary for orthogonality -- unless I miss anything in above and the authors are sure their claim is correct.
> > > >
> > > > # Update
> > > >
> > > > I have read the authors' response "Further clarification on the kernel size" and let me just write an update to my comments here since the thread is already too deep.
> > > >
> > > > After reading the example given by the authors I realized that my point in 4) above needs to be revised: if one is to use cyclic padding on feature map as I proposed for [b] then one also need to do cyclic padding on the kernels. This way the linear convolutional operator in [b] is redefined as the cyclic convolution and all derivations in [b] should carry over. But by having cyclic convolution it is implicitly assuming that the kernel and the signal are of the same size, or in other words, the kernel is zero-padded to be of the same size as the size of the feature map. In conclusion, while one can use cyclic convolution instead of the linear convolution in [b] and their results in Thm 1 should still hold, the correlation operator in the conditions of Thm 1 becomes cyclic correlations with a period that is dependent on the size of the feature map. A similar argument applies to the method in [c]: their kernel may need to be zero-padded according to the feature map size before applying their algorithm.
> > > >
> > > > So now it is clear to me that the author is right, that the notion of orthogonality of cyclic convolutional kernels depends on the input feature map.
> > > >
> > > > I also find the following conjecture from the authors to be very interesting:
> > > >
> > > > > we conjecture that there might exist orthogonal convolutions which can preserve the orthogonality for the input of any size. But these orthogonal convolutions for any size only constitute a subset of orthogonal convolutions for a specific size. Thus, given an input of a specific size, orthogonal convolutions designed for any size might be less complete than orthogonal convolutions for a specific input size.
> > > >
> > > > and I wonder if this conjecture can be proved rigorously by results in [b, c], and perhaps [d - f]. But this will be beyond the scope of this discussion / this submission.
> > > >
> > > > Thanks again for the very detailed response. This is very enlightening discussion. I have no other concerns for the paper.

---

> > > > > ### Author Response · Authors · 2021-11-28
> > > > > **Further clarification on the kernel size (Part 1/3)**
> > > > >
> > > > > Dear Reviewer,
> > > > >
> > > > > We deeply appreciate your detailed comments with great insights.
> > > > > Meanwhile, we are glad to see your positive comments that “ there is nothing technically wrong with the proposed method”, which is really encouraging for us.
> > > > > We also check the paper [b] and [c] again and we agree with you on most of comments. Meanwhile, we still have some different opinions on some small points, which we would like to discuss with you further. Below is our detailed response:
> > > > >
> > > > > **1.	I agree with the authors' comments on [a] that it does not use padding and hence only enforces row or column orthogonality.**
> > > > >
> > > > > Thanks for your time to help us verify this point.
> > > > >
> > > > >
> > > > > **2. I do not agree with the comment that [b] and [a] adopt a similar manner for achieving orthogonality/isometry. [b]   uses zero padding on the feature map (Sec. 2.2, Notations), hence it does not have the issue in [a]. Therefore, [b] derives the notion of orthogonality for convolution (Thm 1) that is independent of the feature map.**
> > > > >
> > > > > We read [b] again and agree with you that [b] adopts the zero padding on the feature map and thus it does not have the issue as [a]. Meanwhile, we also agree with you that the orthogonality for convolution (Thm 1) in [b] is independent of the size of the feature map. Thanks for correcting our misunderstanding on [b].
> > > > >
> > > > > **3. One may argue that this conclusion in [b] cannot possibly hold true since [d] says that orthogonality cannot be achieved with zero padding on the feature map (unless kernel size = 1). My conjecture here is that although Thm 1 in [b] derives equivalent conditions for orthogonality, those conditions cannot hold true unless kernel size = 1.**
> > > > >
> > > > > We agree with you on this point.
> > > > >
> > > > > Due to limited space, we move the response to other points in Part 2 and Part 3.

---

> > > > > ### Author Response · Authors · 2021-11-28
> > > > > **Further clarification on the kernel size (Part 2/3)**
> > > > >
> > > > > **4. The issue of [b] can be fixed if zero padding is replaced by cyclic padding. After applying this fix,  the orthogonality for (cyclic) convolution is independent of the feature map.**
> > > > >
> > > > > To fix the issue in [b], the reviewer proposes to replace  zero padding with cyclic padding. However, the  Thm 1 in [b] is based on zero padding and it might not hold when using cyclic padding.  Below we use a concrete and simple example to show the differences between using zero paddings and using cyclic paddings.
> > > > >
> > > > > To simplify the problem, let’s consider the 1D convolution, and also we assume the number of channels in input, as well as that in the output channel, is 1. Then we define a 1x3 1D convolution kernel [$\alpha$,$\beta$, $\gamma$]. Meanwhile, let’s consider two input vectors of different lengths. The first input vector is  $\mathbf{v}_1 = [x_1, x_2, x_3]$ which is of length 3, and the other input vector is $\mathbf{v}_2 = [x_1, x_2, x_3, x_4]$  which is of length 4.
> > > > >
> > > > > If we adopt the zero padding, when the input is $\mathbf{v}_1$, the Jacobian matrix of convolution,  $\mathbf{J}_1$, is
> > > > >
> > > > > $$\mathbf{J}_1 =
> > > > > \left[\begin{array}{ccc}
> > > > > \alpha & \beta & \gamma \\\\
> > > > > 0 & \alpha & \beta \\\\
> > > > > 0 & 0 & \alpha \\\
> > > > > \end{array}\right]
> > > > > $$
> > > > >
> > > > > When the input is $\mathbf{v}_2$, the Jacobian matrix of convolution,  $\mathbf{J}_2$, is
> > > > >
> > > > >
> > > > > $$\mathbf{J}_2 =
> > > > > \left[\begin{array}{cccc}
> > > > > \alpha & \beta & \gamma & 0 \\\\
> > > > > 0 & \alpha & \beta & \gamma \\\\
> > > > > 0 & 0 & \alpha & \beta  \\\\
> > > > > 0 & 0 & 0 & \alpha
> > > > > \end{array}\right]
> > > > > $$
> > > > >
> > > > >
> > > > > It is not difficult to observe that the orthogonality of $\mathbf{J}_1$ is equivalent to the orthogonality of $\mathbf{J}_2$, that is,
> > > > > \begin{equation}
> > > > > \mathbf{J}_1 \mathbf{J}_1^{\top} = \mathbf{I}  \iff \mathbf{J}_2 \mathbf{J}_2^{\top}= \mathbf{I}.
> > > > > \end{equation}
> > > > > That is, when using zero padding, the orthogonality of a convolution is independent of the size of the input.
> > > > >
> > > > > In contrast, when using cyclic padding, the story changes. To be specific, if we use cyclic padding when the input is $\mathbf{v}_1$, the Jacobian matrix of convolution,  $\mathbf{J}_1$, is
> > > > >
> > > > > $$\mathbf{J}_1 =
> > > > > \left[\begin{array}{ccc}
> > > > > \alpha & \beta & \gamma \\\\
> > > > > \gamma & \alpha & \beta \\\\
> > > > > \beta  & \gamma  & \alpha \\\
> > > > > \end{array}\right]
> > > > > $$
> > > > > To achieve an orthogonal $\mathbf{J}_1$, we should  ensure the below conditions:
> > > > >
> > > > >  1). $[\alpha,\beta, \gamma ] [\gamma, \alpha,\beta]^{\top}  = \alpha\gamma + \beta\alpha + \gamma\beta = 0$,
> > > > >
> > > > >  2). $[\alpha,\beta, \gamma ] [\beta, \gamma, \alpha]^{\top} = \alpha \beta + \beta\gamma + \gamma\alpha =  \alpha\gamma + \beta\alpha + \gamma\beta = 0$,
> > > > >
> > > > > 3). $[\gamma, \alpha,\beta] [\beta, \gamma, \alpha]^{\top} = \gamma\beta + \alpha\gamma + \beta\alpha =  \alpha\gamma + \beta\alpha + \gamma\beta = 0$,
> > > > >
> > > > > 4). $ \alpha^2 + \beta^2 + \gamma^2  =  1$,
> > > > >
> > > > > Note that the first three conditions 1), 2) and 3) can be merged into a single condition $\alpha\gamma + \beta\alpha + \gamma\beta = 0$. Thus, the above four conditions can be merged into below two conditions:
> > > > >
> > > > > 1.a)    $\alpha\gamma + \beta\alpha + \gamma\beta = 0$,
> > > > >
> > > > > 1.b)   $\alpha^2 + \beta^2 + \gamma^2  =  1$,
> > > > >
> > > > >
> > > > >
> > > > > When the input is $\mathbf{v}_2$, the Jacobian matrix of convolution based on cyclic padding,  $\mathbf{J}_2$, is
> > > > > $$\mathbf{J}_2=
> > > > > \left[\begin{array}{cccc}
> > > > > \alpha & \beta & \gamma & 0  \\\\
> > > > > 0 & \alpha & \beta & \gamma   \\\\
> > > > > \gamma  & 0 & \alpha & \beta \\\\
> > > > > \beta & \gamma  & 0 & \alpha
> > > > > \end{array}\right]
> > > > > $$
> > > > >
> > > > >
> > > > >
> > > > >
> > > > > To achieve an orthogonal $\mathbf{J}_2$, we should ensure
> > > > >
> > > > > 1).$[\alpha, \beta, \gamma, 0 ]  [0,  \alpha, \beta, \gamma]^{\top} = \beta(\alpha+\gamma)  = 0$,
> > > > >
> > > > > 2). $[\alpha, \beta, \gamma, 0  ] [\gamma, 0, \alpha, \beta]^{\top} = 2\alpha\gamma = 0$,
> > > > >
> > > > >  3). $[\alpha, \beta, \gamma, 0  ]  [\beta, \gamma, 0, \alpha ]^{\top} = \beta(\alpha+\gamma)  = 0$,
> > > > >
> > > > > 4). $  [0,  \alpha, \beta, \gamma] [\gamma, 0, \alpha, \beta]^{\top} =   \beta(\alpha+\gamma)  = 0$,
> > > > >
> > > > > 5). $ [0,  \alpha, \beta, \gamma] [\beta, \gamma, 0, \alpha ]^{\top} =  2\alpha\gamma = 0$,
> > > > >
> > > > > 6). $ [\gamma, 0, \alpha, \beta] [\beta, \gamma, 0, \alpha ]^{\top} =   \beta(\alpha+\gamma)  = 0$,
> > > > >
> > > > > 7). $ \alpha^2 + \beta^2 + \gamma^2  =  1$.
> > > > >
> > > > > The above 7 conditions can be merged into below three conditions:
> > > > >
> > > > > 2.a) $\beta(\alpha+\gamma)= 0$,
> > > > >
> > > > > 2.b) $2\alpha\gamma = 0$,
> > > > >
> > > > > 2.c) $ \alpha^2 + \beta^2 + \gamma^2  =  1$.
> > > > >
> > > > >
> > > > >
> > > > > Based on the above analysis, the condition for achieving orthogonal $\mathbf{J}_1$ (1.a and 1.b) is no longer equivalent to achieving orthogonal $\mathbf{J}_2$  (2.a, 2.b and 2.c) when using cyclic padding, that is
> > > > >
> > > > > \begin{equation}
> > > > > \mathbf{J}_1 \mathbf{J}_1^{\top} = \mathbf{I}  \nLeftrightarrow \mathbf{J}_2 \mathbf{J}_2^{\top}= \mathbf{I}.
> > > > > \end{equation}
> > > > > Due the limited space, we move the proof of $\mathbf{J}_1 \mathbf{J}_1^{\top} = \mathbf{I}  \nLeftrightarrow \mathbf{J}_2 \mathbf{J}_2^{\top}= \mathbf{I}$ to part 3 of the response.
> > > > > That is, the orthogonality of a convolution might be dependent on the size of input feature.

---

> > > > > ### Author Response · Authors · 2021-11-28
> > > > > **Further clarification on the kernel size (Part 3/3)**
> > > > >
> > > > > **The proof of  $\mathbf{J}_1 \mathbf{J}_1^{\top} = \mathbf{I}  \nLeftrightarrow \mathbf{J}_2 \mathbf{J}_2^{\top}= \mathbf{I}$**
> > > > >
> > > > > Recall  that the condition for $\mathbf{J}_1 \mathbf{J}_1^{\top} = \mathbf{I} is:
> > > > >
> > > > > 1.a)    $\alpha\gamma + \beta\alpha + \gamma\beta = 0$,
> > > > >
> > > > > 1.b)   $\alpha^2 + \beta^2 + \gamma^2  =  1$.
> > > > >
> > > > > To obtain a solution satisfying 1.a  and 1.b, we devise
> > > > >
> > > > > $\alpha = sin(\phi)$ ,  $\beta = cos(\phi) sin(\psi)$, $\gamma= cos(\phi) cos(\psi)$.
> > > > > It is not diffucult to observe that they naturally satisfy the condition 1.b), then we plug them in 1.a) and obtain:
> > > > >
> > > > > \begin{equation}
> > > > > sin(\phi) cos(\phi) cos(\psi) + cos(\phi) sin(\psi)sin(\phi)   + cos(\phi) cos(\psi)cos(\phi) sin(\psi) = 0,
> > > > > \end{equation}
> > > > > which means  $cos(\phi)=0$, or
> > > > > \begin{equation}
> > > > > sin(\phi)  cos(\psi) +  sin(\psi)sin(\phi)   +  cos(\psi)cos(\phi) sin(\psi) = 0,
> > > > > \end{equation}
> > > > >
> > > > > which is equivalent to
> > > > > \begin{equation}
> > > > > tan(\phi)  cos(\psi) +  sin(\psi)tan(\phi)   +  cos(\psi)sin(\psi) = 0,
> > > > > \end{equation}
> > > > >
> > > > > which can be further rearranged into:
> > > > > \begin{equation}
> > > > > tan(\phi)  (cos(\psi) +  sin(\psi))  =  -cos(\psi)sin(\psi).
> > > > > \end{equation}
> > > > > Let we choose $\psi = \pi/4$,  $tan(\phi)  = \frac{-cos(\psi)sin(\psi)}{cos(\psi) +  sin(\psi)} = -1/2\sqrt{2}$, $cos(\phi) = \sqrt{\frac{1}{1+tan^2(\phi)}} = 2\sqrt{2}/3$ and $sin(\phi) = cos(\phi) tan(\phi)= -1/3$.
> > > > > Then $\alpha = sin(\phi) = -1/3$, $\beta = cos(\phi) sin(\psi) =2/3 $, and $\gamma= cos(\phi) cos(\psi) = 2/3$.
> > > > >
> > > > > We can plug $(\alpha = -1/3, \beta =2/3,\gamma= 2/3 )$ into condition 1.a and 1.b, it is not difficult to observe that
> > > > >  condition 1.a and 1.b hold.
> > > > >
> > > > > Recall  that the condition for $\mathbf{J}_2 \mathbf{J}_2^{\top} = \mathbf{I}$ is:
> > > > >
> > > > > 2.a) $\beta(\alpha+\gamma)= 0$,
> > > > >
> > > > > 2.b) $2\alpha\gamma = 0$,
> > > > >
> > > > > 2.c) $ \alpha^2 + \beta^2 + \gamma^2  =  1$.
> > > > > It is obvious that $(\alpha = -1/3, \beta =2/3,\gamma= 2/3 )$ do not satisfy condition  2.a), 2.b)  and 2.c). Thus we finish our proof.
> > > > >
> > > > > Meanwhile, based on the above analysis, we can obtain an insteresting observation:
> > > > > \begin{equation}
> > > > >  \mathbf{J}_2 \mathbf{J}_2^{\top} = \mathbf{I}  \Rightarrow \mathbf{J}_1 \mathbf{J}_1^{\top} = \mathbf{I} .
> > > > > \end{equation}
> > > > > It can be easily proven by plugging the conditions 2.a, 2.b and 2.c into conditions 1.a and 1.b.
> > > > >
> > > > >
> > > > >
> > > > > This means that an orthogonal convolution with the kernel of size 3 for the input of length 4 can naturally achieve the orthogonality for the input of length 3. But the reverse does not hold. That is, the set of all orthogonal convolution layers with a kernel of size 3 for the input of length 4 is just a subset of that with a kernel of size 3 for the input of length 3.
> > > > >
> > > > >
> > > > >
> > > > >
> > > > >
> > > > > Based on the above analysis, we conjecture that there might exist orthogonal convolutions which can preserve the orthogonality for the input of any size. But these orthogonal convolutions for any size only constitute a subset of orthogonal convolutions for a specific size.  Thus,  given an input of a specific size, orthogonal convolutions designed for any size might be less complete than orthogonal convolutions for a specific input size.
> > > > >
> > > > >
> > > > >
> > > > > **5. Moving to [c], I do not agree with the statement that it ignores padding. The proof to Theorem A.5 explicitly states that they do circular padding for finite inputs. Hence, [c] is another example where the notion of orthogonality is independent of kernel size.**
> > > > >
> > > > > We check [c] again and agree with you that they also use circular/cyclic padding.  We think  [c]  constructs the orthogonal convolution which can preserve the orthogonality for  any-size input.  Based on our analysis above, the constructed orthogonal convolutions constitute a subset of the orthogonal convolutions for a specific input size. When the input is of a specific size,  the orthogonal convolution designed for any-size input might be less complete than the orthogonal convolution designed for a specific input size.

---

> > > > > ### Author Response · Authors · 2021-11-29
> > > > > **The Response to the Update from Reviewer YtM2**
> > > > >
> > > > > Dear Reviewer YtM2,
> > > > >
> > > > > We are grateful for your very inspiring comments, which indeed inspire us a lot and motivate us to think more deeply into the problem.
> > > > > We also learn a lot during the discussion and feel pleasant to clarify some important issues which we have never thought about before.
> > > > > Last but not the least, we thank you for raising the score from 5 to 8.  The positive feedback from you is indeed an encouragement for us.
> > > > >
> > > > > Sincerely,
> > > > > Authors

---

### Official Review · Reviewer_6wTV · 2021-11-03

**Correctness:** 4
**Technical Novelty And Significance:** 3
**Empirical Novelty And Significance:** 3
**Recommendation:** 6
**Confidence:** 3

**Main Review:**

Pro:
The authors propose a theoretically-motivated way of constructing orthogonal convolutions that achieves good robust performance on several benchmarks.

It has a big advantage on the testing speed over the previous state-of-the-art, SOC. I think this is an important step in this research direction.

Cons:
The proposed method is still limited to the 1-Lipschitz architecture. What is the challenge to extend the proposed method to more advanced architectures?

Comments:

In my opinion, it will be helpful to have another entry showing how ordinary convolution performs in table 2 and table 4. It may happen that the ordinary convolution will have higher standard accuracy, but by showing they are not robust at all, the advantage of the proposed method will be further highlighted.

Another line of enforcing orthogonal constraint in convolutional neural networks should also be discussed:
[1] orthogonal convolutional neural networks. CVPR 2020
[2] deep isometric learning for visual recognition. ICML 2020.

**Summary Of The Paper:**

This paper studies how to construct orthogonal convolutional networks in an efficient way. To this end, this paper builds the connection between the DFT-transformed kernel with the common dilated convolution. During training, the forward pass can be done by a sequence of inverse DFT and dilated convolution. During testing, all the convolution kernels only need to be transformed once so that evaluation time is significantly reduced.

**Summary Of The Review:**

I think this paper contributes a new idea in this field.

---

> ### Author Response · Authors · 2021-11-12
> **Initial response to Reviewer 6wTV**
>
> We thank the reviewer for helpful comments and encouraging rating. In this initial response, we clarify the issues raised by the reviewer to receive more feedback in time. Please look forward to our revised manuscript in next few days, and we also look forward to your feeback after you read our initial response.
>
> **Q1: In my opinion, it will be helpful to have another entry showing how ordinary convolution performs in table 2 and table 4. It may happen that the ordinary convolution will have higher standard accuracy, but by showing they are not robust at all, the advantage of the proposed method will be further highlighted.**
>
> A1: Thanks for your constructive suggestions. We will add the experimental results of the ordinary convolution to demonstrate the advantage of the orthogonal convolution in robust accuracy.
>
> **Q2: Another line of enforcing orthogonal constraint in convolutional neural networks should also be discussed: [1] orthogonal convolutional neural networks. CVPR 2020 [2] deep isometric learning for visual recognition. ICML 2020.**
>
> A2: Thanks for your suggestions. We will add the discussions on these two works in the revised manuscript.  In fact, these two works encourage the orthogonality of the Jacobian matrix of convolution layers through a regularizer, thus,  they can not achieve the  strict orthogonality and the provable robustnesss.

---

> > ### Comment · Reviewer_6wTV · 2021-11-20
> > **RE: Initial response to Reviewer 6wTV**
> >
> > Thanks for your reply. My concerns are well addressed.

---

> > > ### Author Response · Authors · 2021-11-21
> > > **Response to Reviewer 6wTV's Response**
> > >
> > > Dear Reviewer 6wTV,
> > >
> > > We thank again for your efforts in helping improve the quality of the submitted manuscript and we are glad that we have addressed your concerns. We appreciate your kind and positive feedback, which is indeed encouraging for us.
> > >
> > > Best,
> > >
> > > Anonymous Authors

---

### Official Review · Reviewer_mniG · 2021-11-03

**Correctness:** 3
**Technical Novelty And Significance:** 3
**Empirical Novelty And Significance:** 2
**Recommendation:** 3
**Confidence:** 5

**Main Review:**

Overall, I think the idea in the paper is novel and elegant. However, there are a few aspects in the current draft that are not carefully developed/explained.

1) The limitations of the proposed method. a) In previous approaches (BCOP, Cayley, SOC), the convolution kernel is orthogonal for ANY resolution. In contrast, the convolution kernel is orthogonal ONLY IF the resolution equals the product of filter size and dilation. b) As a result, the method limits the flexibilities of orthogonal convolutions --- for example, the method can not construct orthogonal convolutions for features with a prime-number resolution. While orthogonal convolutions are rarely used for different resolutions or prime-number resolution, I think the authors shall discuss these two limitations explicitly.

2) Real frequency domain. The matrix P is the Fourier transform of the convolution kernel, which generally is complex-valued. Restricting to the real-valued matrix has two drawbacks. a) The filters must be even, i.e., the method cannot construct orthogonal convolutions that are asymmetric. b) The space of real orthogonal matrices is disconnected, and the matrix exponential can only cover one component SO(c). I don't see why the authors do not adopt unitary matrices, which do not suffer from these two drawbacks.

3) Computation of matrix exponential. The paper uses truncated Taylor expansion instead of other more computation/memory efficient methods --- The SOC paper adopts truncated Taylor series because there is no efficient algorithm when A is a convolution instead of a matrix. Indeed, a cited article (Casado & Martínez-Rubio 2019) provides an exact and efficient algorithm to parameterize orthogonal/unitary matrices. However, using the current method, the authors will have to analyze how truncation error affects the exact orthogonality of the convolution kernel.

4) Other minor issues. a) At the end of the second paragraph of the Introduction (Section 1), the authors write "Straightforwardly, we can ...". This is not correct. Even one can expand the kernel to a Jacobian matrix, it is not straightforward to construct a matrix that is both block-Toeplitz and orthogonal. b) In the second paragraph of the Implementation (Section 5), I think the padding will be proportional to the product of filter size and dilation instead of addition. c) The experiments only repeat the ones in SOC paper, which do not fully demonstrate the proposed method in other settings (e.g., Wasserstein estimation).

**Summary Of The Paper:**

The paper proposes an economical method to construct orthogonal convolutions using dilated convolutions.

**Summary Of The Review:**

In summary, I believe the idea in the paper is novel and promising. However, the authors have not thoroughly/carefully developed the picture (as explained above). Therefore, I suggest rejecting the paper for now --- but I look forward to upcoming revisions.

---

> ### Author Response · Authors · 2021-11-12
> **The initial response to Reviewer mniG**
>
> We are grateful for  constructive comments from the reviewer, which are indeed helpful for improving the quality of the submitted manuscript. In order to obtain more feedback from the reviewer, below, we clarify some issues raised by the reviewer to facilitate understanding the details of the proposed method. Meanwhile, we are revising the manuscript and will update it in the next few days. We look forward to more replies after the reviewer reads our response.
>
> **Q1: The limitations of the proposed method are two-fold: a). In previous approaches (BCOP, Cayley, SOC), the convolution kernel is orthogonal for ANY resolution. In contrast, the convolution kernel is orthogonal ONLY IF the resolution equals the product of filter size and dilation. b). The method can not construct orthogonal convolutions for features with a prime-number resolution, although orthogonal convolutions are rarely used for different resolutions or prime-number resolution.**
>
>
> A1: As for point a), we agree with the reviewer on the comments on BCOP and SOC that their convolution kernel can process feature maps of ANY resolution. But for Carley (Asher Trockman et al. ICLR’21), its convolution kernel must be in the same size as the feature map. In summary, ours is more flexible than Carley and less flexible than BCOP and SOC. As for the point b), given a feature map of size $n\times n$ where n is a prime, we can always use zero padding to increase the size to $(n+p)\times (n+p)$ where $n+p$ is divisible by the convolution kernel size $k$.
>
> **Q2:   The matrix P is the Fourier transform of the convolution kernel, which generally is complex-valued. Restricting to the real-valued matrix has two drawbacks. a) The filters must be even, i.e., the method cannot construct orthogonal convolutions that are asymmetric. b) The space of real orthogonal matrices is disconnected, and the matrix exponential can only cover one component SO(c). I don't see why the authors do not adopt unitary matrices, which do not suffer from these two drawbacks.**
>
> A2: We agree with the reviewer’s comments on the limitations of real value in the frequency domain. In fact, In the paragraph above Table 1 of the originally submitted manuscript, we also claim that P is not necessarily real and a complex P can make the constructed orthogonal convolution kernel more complete.  Meanwhile,  in the originally submitted manuscript, we have already compared the real P  and the complex P, which is unitary  in Appendix B.  The complex setting achieves higher recognition accuracy but takes much more training time.
>
>
> **Q3: Computation of matrix exponential. The paper uses truncated Taylor expansion instead of other more efficient methods --- The SOC paper adopts truncated Taylor series because there is no efficient algorithm when A is a convolution instead of a matrix. Indeed, a cited article (Casado & Martínez-Rubio 2019) provides an exact and efficient algorithm to parameterize orthogonal/unitary matrices. However, using the current method, the authors will have to analyze how truncation error affects the exact orthogonality of the convolution kernel.**
>
> A3:  Thanks for suggestions. The purpose of using the truncated Taylor expansion instead of other more computation/memory efficient methods is to make the comparison between our method and SOC fair. If we use more efficient methods for orthogonalizing weights, then the high efficiency of our method might be contributed to a more advanced weight orthogonalization approach. In the pointed work (Casado & Martínez-Rubio 2019), the weight orthogonalization is achieved through matrix exponential.  To compute the exact matrix exponential in (Casado & Martínez-Rubio 2019), we need SVD, which is not friendly for parallelism and thus is slow in GPU planform. To boost the efficiency, SOC and ours approximate matrix exponential through Taylor expansions.  In the revised manuscript. We will add the experimental results to show the orthogonality of P with respect to the truncation error.
>
> **Q4: At the end of the second paragraph of the Introduction (Section 1), the authors write, "Straightforwardly, we can ...".  This is not correct. Even one can expand the kernel to a Jacobian matrix. It is not straightforward to construct a matrix that is both block-Toeplitz and orthogonal.**
>
> A4: Thanks for your comments. We agree with your comments and will fix the statement. Constructing a block-Toeplitz and orthogonal Jacobian matrix is indeed not straightforward and much harder than what we previously expected.
>
> **Q5:  In the second paragraph of the Implementation (Section 5), I think the padding will be proportional to the product of filter size and dilation instead of addition.**
>
> A5: Yes, it should be the product of filter size and dilation. We will fix it in the revised manuscript. To be specific, given a convolution of the kernel size k and the dilation d, and the stride 1, the padding is d(k-1) on the single side or d(k-1)/2 on double sides.

---

> > ### Comment · Reviewer_mniG · 2021-11-20
> > **Response to the authors**
> >
> > Dear authors,
> >
> > Thank you for your detailed responses! I have some follow-up questions/concerns about your answers.
> >
> > A1: For CayleyConv, I think the underlying kernel size is independent of the feature size. So the claim for CayleyConv is not valid.
> >
> > A2: I wonder if it is possible to move the case that P is complex-valued to the main paper and consider that P is real-valued as an ablation study. Using a more general/expressive/accurate case as an ablation study in the appendix is not common.
> >
> > A3: I think the matrix exponential in (Casado & Martínez-Rubio 2019) does not rely on SVD --- it uses the scaling and squaring method by Higman [1]. I am also surprised that the truncated Taylor expansion is even faster than the method in (Casado & Martínez-Rubio 2019). If this is the case, what is the need to develop efficient methods in (Casado & Martínez-Rubio 2019)? It is counter-intuitive as the line of works [2] for matrix exponential aims to accelerate matrix exponential compared to simple Taylor expansion. Could the authors provide more analysis of why this is the case?
> >
> > [1] Higham, Nicholas J. "The scaling and squaring method for the matrix exponential revisited." SIAM Journal on Matrix Analysis and Applications 26.4 (2005): 1179-1193.
> >
> > [2] Moler, Cleve, and Charles Van Loan. "Nineteen dubious ways to compute the exponential of a matrix, twenty-five years later." SIAM Review 45.1 (2003): 3-49.
> >
> > Best,
> >
> > Reviewer mniG

---

> > > ### Author Response · Authors · 2021-11-21
> > > **The Response to Reviewer mniG's Response**
> > >
> > > Dear Reviewer mniG,
> > >
> > > We deeply appreciate your insightful response, which further helps us understand the problem. Below we make further clarifications and look forward to more feedback from you to further help us improve the manuscript.
> > >
> > > **Q1:  For CayleyConv, I think the underlying kernel size is independent of the feature size. So the claim for CayleyConv is not valid.**
> > >
> > > A1: We have double checked CayleyConv  to ensure that our claim is valid.  As stated in the second paragraph  and the third paragraph of Section 4 in CayleyConv,  the shape of the convolution weight is of shape $\mathbb{R}^{c_{out}\times c_{in}\times n \times n}$  and the input $\mathbf{X}$ is of shape  $\mathbb{R}^{c_{in}\times n \times n}$. We believe that this setting requires the spatial size of the weight kernel and that of the input feature map to be equal. Meanwhile, please also refer to Algorithm 1 in CayleyConv, for each $i,j \in [1,n] \times [1,n]$, it processes the  Fourier Transform of the input, $\hat{\mathbf{X}}[:,:,i,j]$, based on the Fourier Transform of the weight, $\hat{\mathbf{W}}[:,:,i,j]$. If the spatial size of $\mathbf{W}$ is not equal to $n\times n$,  it will be impossible to conduct this step. Could you please also kindly help us double check the second paragraph  and the third paragraph of Section 4 in CayleyConv  and  Algorithm 1 in CayleyConv to see if our understanding is correct or not?  We also want to make this point clear so that we will not make mistakes on this point.
> > >
> > >
> > > **Q2:   I wonder if it is possible to move the case that P is complex-valued to the main paper and consider that P is real-valued as an ablation study. Using a more general/expressive/accurate case as an ablation study in the appendix is not common.**
> > >
> > > A2:  Thanks for your suggestions. We also consider the case that P is complex-valued to the main paper and consider that P is real-valued as an ablation study. But our concern is that, the complex-value P is very slow in training, making our method not as efficient as the counterpart SOC in the training phase, which might contradict our initial motivation for improving the efficiency of the current orthogonal convolution network. We are still thinking about it.
> > >
> > >
> > > **Q3:  I think the matrix exponential in (Casado & Martínez-Rubio 2019) does not rely on SVD --- it uses the scaling and squaring method by Higman [1]. I am also surprised that the truncated Taylor expansion is even faster than the method in (Casado & Martínez-Rubio 2019). If this is the case, what is the need to develop efficient methods in (Casado & Martínez-Rubio 2019)? It is counter-intuitive as the line of works [2] for matrix exponential aims to accelerate matrix exponential compared to simple Taylor expansion. Could the authors provide more analysis of why this is the case?**
> > >
> > > A3:   Thanks for correcting our misunderstanding in  (Casado & Martínez-Rubio 2019).  We have double checked  (Casado & Martínez-Rubio 2019)  and agree with your that (Casado & Martínez-Rubio 2019) do not rely on SVD.  Instead they use the scaling and squaring method in [a] (as mentioned in Section 4.2 in [a]). Meanwhile, we also agree with you that the scaling and squaring method in [a] should be a more efficient method than a simple Taylor expansion in generally.   The time cost for matrix exponential in our previous response (12.8 ms) is based on pytorch's implementation on matrix exponential function, torch.matrix_exp, and that's why it is slower than our Taylor expansion. We believe that if we use the more advanced matrix exponential approximation method in [a], our method can be faster.  In fact, our current implementation based on Taylor expansion has been very fast (it only takes 5 iterations to achieve a good approximation). Specifically,  it only takes 2.5 ms per layer, which has already occupied small portion in the whole training time.  The efficiency improvement brought by using more advanced approximation in [a] might be marginal.
> > >
> > >
> > > | matrix exponential (torch.matrix_exp) | truncated Taylor expansion |
> > > |--------------------|----------------------------|
> > > | 12.8ms             | 2.5ms                      |
> > >
> > > [a] Al-Mohy, A. H. and Higham, N. J. A new scaling and squaring algorithm for the matrix exponential. SIAM Journal on Matrix Analysis and Applications, 31(3):970– 989, 2009b
> > >
> > >
> > > Best,
> > >
> > > Anonymous Authors

---

> > > > ### Author Response · Authors · 2021-11-22
> > > > **Additional Response to the Response from mniG**
> > > >
> > > > Hi, Dear Reviewer mniG
> > > >
> > > > Again, we deeply appreciate your kind help in improving the submitted manuscript.
> > > > Since the discussion stage deadline is approaching, we wonder if we have addressed your concerns and questions in the response?  If any issues remain, we are still be active for addressing them.
> > > >
> > > > Best,
> > > >
> > > > Anonymous Authors

---

> > > > > ### Comment · Reviewer_mniG · 2021-11-29
> > > > > **Response to the authors**
> > > > >
> > > > > Dear authors,
> > > > >
> > > > > Thanks for your response. Here are follow-up comments on your previous replies.
> > > > >
> > > > > 1. For CayleyConv, the $k \times k$ kernel is first padded to the input size $n \times n$ (See **Runtime comparison** in Section 4) --- the underlying kernel size $k$ is independent to the input size $n$. It is not hard to verify that the CayleyConv layer is orthogonal for any $n$ (not only when $k = n$). Therefore, CayleyConv is as flexible as other approaches (BCOP and SOC).
> > > > >
> > > > > 2. I think efficiency is not the priority before a method becomes mature; 3. Since there are many methods for the matrix exponential (using advanced algorithms or parallel computing), it is possible to make the proposed design faster than SOC. When the authors use efficiency as a primary motivation, the current study (only trying one naive approach) is far from enough.
> > > > >
> > > > > While I appreciate the novelty of the proposed design, I still think the current draft is not fully ready for publication (due to a lack of comprehensive understanding of previous works and insufficient investigation of efficient algorithms).
> > > > >
> > > > > Best,
> > > > >
> > > > > Reviewer mniG

---

> > > > > > ### Author Response · Authors · 2021-11-29
> > > > > > **Response to Reviewer mniG's new response**
> > > > > >
> > > > > > Dear Reviewer mniG,
> > > > > >
> > > > > > Thanks so much for your timely reply, which gives us enough time for further clarifying the unclear issues. We look forward to more feedback from you.
> > > > > >
> > > > > > **Q1. For CayleyConv, the $k\times k$ kernel is first padded to the input size $n\times n$ (see runtime comparison in Section 4). The underlining kernel size $k$ is independent of the input size.  It is not hard to verify that the CayleyConv layer is orthogonal for any n (not only when k=n). Therefore, CayleyConv is as flexible as other approaches (BCOP and SOC).**
> > > > > >
> > > > > > Thanks for making the problem more clearly and we get your point.  We agree with you that CayleyConv first zero-pads $k\times k$ convolution kernel to $n\times n$, and $k$ is not necessarily equal to $n$. If it directly uses the zero-padded $n\times n$ convolution kernel for convolution, the receptive field will still be of  $k \times k$ size, and CayleyConv will support the convolution of any spatial size. However,  as shown in Algorithm 1 of CayleyConv, it conducts Cayley transform on convolution kernel in the frequency domain, which increases the size of receptive field from $k\times k$ to $n \times n$. That is, no matter how you choose $k$, the receptive field of  CayleyConv is always $n\times n$.  To make it more clear, below let us look into the detailed process of  CayleyConv in Algorithm 1.
> > > > > > As shown in Algorithm 1, CayleyConv consists of  three steps as shown below:
> > > > > >
> > > > > > **(1) FFT on both weights $W$ to obtain the frequency domain components $\hat{W}$ and FFT on the feature map $X$ to obtain the frequency domain components $\hat{X}$ (line 1 in Algorithm 1).**
> > > > > >
> > > > > > $$
> > > > > > \hat{W} = \mathrm{FFT}({W} )
> > > > > > $$
> > > > > >
> > > > > > We assume $W$ is obtained from zero padding a $k\times k \times c_{in} \times c_{out} $ convolution to $n \times n \times c_{in} \times c_{out}$, and $k < n$. That is, the size of the receptive field of the initial convolution kernel $W$ is only $k\times k$ instead of $n\times n$.
> > > > > >
> > > > > >
> > > > > >
> > > > > >
> > > > > >
> > > > > > **(2) Cayley Transform on $\hat{W}$ followed by multiplying $\hat{X}$ to obtain $\hat{Z}$ (line 4-6 in Algorithm 1).**
> > > > > >
> > > > > > $$
> > > > > > \hat{A} = \hat{W} - \hat{W^*}
> > > > > > $$
> > > > > >
> > > > > >
> > > > > > $$
> > > > > > \hat{Y} = (I + \hat{A})^{-1} \hat{X}
> > > > > > $$
> > > > > >
> > > > > >
> > > > > > $$
> > > > > > \hat{Z} =  \hat{Y} - \hat{A}\hat{Y}
> > > > > > $$
> > > > > > For the easiness of illustration,  the above three lines are merged into a single line:
> > > > > >
> > > > > > $$
> > > > > > \hat{Z} =  [(I -  \hat{W} + \hat{W^*} ) (I +  \hat{W} - \hat{W^*})^{-1} ]  \hat{X}.
> > > > > > $$
> > > > > > We denote the Cayley function by $C(\hat{W}) = (I -  \hat{W} + \hat{W^*} ) (I +  \hat{W} - \hat{W^*})^{-1}$,  the above equation can be rewritten as
> > > > > >
> > > > > > $$
> > > > > > \hat{Z} =  C(\hat{W}) \hat{X}.
> > > > > > $$
> > > > > >
> > > > > >
> > > > > > **(3) Inverse FFT on the frequency domain $\hat{Z}$ to get back the output feature map $Z$ in the spatial domain  (line 8 in Algorithm 1 ).**
> > > > > >
> > > > > > $$
> > > > > > Z = \mathrm{FFT}^{-1}(\hat{Z})
> > > > > > $$
> > > > > >
> > > > > > As $\hat{Z} =  C(\hat{W}) \hat{X}$, we have
> > > > > > $$
> > > > > > Z = \mathrm{FFT}^{-1}(\hat{Z}) =  \mathrm{FFT}^{-1}( C(\hat{W}) \hat{X}) =   \mathrm{FFT}^{-1}(C(\hat{W})) \circledast \mathrm{FFT}^{-1}((\hat{X})) =  \mathrm{FFT}^{-1}(C(\hat{W})) \circledast X,
> > > > > > $$
> > > > > > where $\circledast$ denotes the convolution operation. The above equation is based on the convolution theorem
> > > > > > $\mathrm{FFT}^{-1}(\mathrm{FFT}(X)\mathrm{FFT}(Y)) = X \circledast Y $ and $\mathrm{FFT}^{-1}(\mathrm{FFT}(X)) = X$.
> > > > > >
> > > > > >
> > > > > > That is, in CayleyConv, the factual convolution kernel is   $\mathrm{FFT}^{-1}(C(\hat{W}))$ instead of $W$.
> > > > > > Even though $W$ has a local $k\times k$ receptive field ( $W$ contains only $k\times k \times c_{in} \times c_{out}$ non-zero elements as it is padded from a $k\times k$ kernel),  $\mathrm{FFT}^{-1}(C(\hat{W}))$ has a global $n \times n$ receptive field ( $n\times n \times c_{in} \times c_{out}$ non-zero elements). **It explains that, in Runtime comparison of Section 4, the complexity is only related with $n$ and is irrelevant with $k$.**
> > > > > > So we still argue that, in CayleyConv, the convolution kernel  must be of the same size as input feature map. Thus it is not flexible as SOC which can use the kernel of any size.  Notably,  the convolution kernel we mention here is not $W$ but the factual kernel $\mathrm{FFT}^{-1}(C(\hat{W}))$.
> > > > > >
> > > > > >
> > > > > > **Q2: I think efficiency is not the priority before a method becomes mature.**
> > > > > >
> > > > > > We agree with you on this point. Since we have already conducted the experiments based on the complex settings which we have presented it in the Appendix, we think it is not a big issue to move it to the main manuscript.
> > > > > >
> > > > > >
> > > > > >
> > > > > > **Q3. Since there are many methods for the matrix exponential (using advanced algorithms or parallel computing), it is possible to make the proposed design faster than SOC.**
> > > > > >
> > > > > > We agree with you on this point. Our framework can more flexibly support more manners for the matrix exponential and thus can be  faster. However, we would like to highlight that, the main contribution of this work is to constructing orthogonal convolution in an explicit manner. Matrix exponential is only one step in our method and we can feel free to choose any method for the matrix exponential to achieve our goal.
> > > > > >
> > > > > > Sincerely,
> > > > > >
> > > > > > Authors

---

> > > > > > > ### Author Response · Authors · 2021-11-29
> > > > > > > **Additional Analysis on Flexibility of Existing Methods to Reviewer mniG**
> > > > > > >
> > > > > > > Dear Reviewer mniG,
> > > > > > >
> > > > > > > We highly appreciate your insightful comments, which motivate us to rethink the limitations of our work and the existing methods. Below we would like to discuss more on the limitations of existing methods.
> > > > > > >
> > > > > > > **As mentioned by the Reviewer mniG, the existing methods such as CarleyConv [1] and SOC [2] could support convolution of any spatial size. In the above response, we have analysed that the CarleyConv is not as flexible as Reviewer mniG thought. Now we find the size of the receptive field of SOC can not be any k, either. Below we give our detailed analysis.**
> > > > > > >
> > > > > > >
> > > > > > > Let us focus on Equation (5) in  SOC [2], it computes the convolution in the below way:
> > > > > > >
> > > > > > > \begin{equation}
> > > > > > > \mathbf{L}_e \star \mathbf{X} = \mathbf{X} + \mathbf{L}  \star \mathbf{X} + \frac{ \mathbf{L} \star^2 \mathbf{X}}{2!} + \frac{ \mathbf{L} \star^3 \mathbf{X}}{3!} + \cdots + \frac{ \mathbf{L} \star^T \mathbf{X}}{T!} ,
> > > > > > > \end{equation}
> > > > > > >
> > > > > > > where $\mathbf{L}$ is a convolutional kernel and $\star$ denotes the convolution operation. Notably, the convolution kernel $\mathbf{L}$ is of size $k\times k \times c_{in} \times c_{out}$ and $k$ can be any value.  If we only look at $\mathbf{L}$, it seems that the SOC can support the convolution with the receptive field of any size.  However,  the real convolution kernel is $\mathbf{L}_e$ in the above equation instead of $\mathbf{L}$.  As shown in  the above equation,  $\mathbf{L}_e \star \mathbf{X}$ is a summation of $T+1$ items. Let us focus on the last item   $\frac{ \mathbf{L} \star^T \mathbf{X}}{T!}$.  It iteratively conducts $T$ times convolution using the convolution kernel $\mathbf{L}$, which is equivalent to stacking $T$ identical convolution layers of kernel  $\mathbf{L}$. It is worth noting that, if we stack $T$
> > > > > > > kernels and each kernel is with the receptive field of $k \times k$ size, the receptive field of the stack of convolution layers will increase from $k \times k$  to $(k +(T-1)(k-1)) \times (k +(T-1)(k-1))$.  Thus,  if we use a $k\times k$ convolution kernel $\mathbf{L}$, the factual receptive field of SOC is $(k +(T-1)(k-1)) \times (k +(T-1)(k-1))$.
> > > > > > >
> > > > > > > Stacking a few convolution layers of the small receptive field to achieve a large-scale receptive field is a widely used trick in building efficient convolution. For example, in a famous early work [3], the authors stack two $3\times 3$ convolution layers  to achieve a convolution of $5 \times 5$ receptive field.
> > > > > > >
> > > > > > >
> > > > > > > [1] Trockman, Asher, and J. Zico Kolter. "Orthogonalizing convolutional layers with the Cayley transform." ICLR (2021)
> > > > > > >
> > > > > > > [2] Singla, Sahil, and Soheil Feizi. "Skew Orthogonal Convolutions." ICML (2021).
> > > > > > >
> > > > > > > [3] Szegedy, Christian, et al. "Rethinking the inception architecture for computer vision." CVPR (2016).

---

> > > > > > > > ### Comment · Reviewer_mniG · 2021-11-30
> > > > > > > > **Clarification of our question**
> > > > > > > >
> > > > > > > > I think the authors have misunderstood some of my comments.
> > > > > > > > - For flexibility, I mean the hyperparameter $k$ that determines the filter size is independent of the feature size $n$ --- one need not know $n$ beforehand when choosing $k$. For example, one can train a convolutional network on $n \times n$ inputs but test it for $n^\prime \times n^\prime$ (with $n^\prime = n$). However, this proposed method does not have this flexibility.
> > > > > > > > - Though not directly related to my comment, I would like to point out a misunderstanding --- the authors confuse **the filter** with **the algorithm to perform filtering**. In digital signal processing, the filter in BCOP is *finite impulse response (FIR)* (with finite receptive field). In contrast, the filter in CayleyConv/SOC is *infinite impulse response (IIR)* (with possible infinite receptive field). Since the inputs are circular-padded (and thus have an infinite length in math), an IIR filter applies on the infinite-dimensional space, where all input elements contribute to each output element. However, the notion of filter shall not be confused with the algorithm that performs the filtering operation. For example, one can also use FFT to compute a standard convolutional layer with kernel size $k \times k$ -- the algorithm first zero-pads the kernel to $n \times n$, performs Fourier transform on $n \times n$, etc. However, it does not make sense to say the algorithm increases kernel size from $k \times k$ to $n \times n$.

---

> > > > > > > > > ### Author Response · Authors · 2021-11-30
> > > > > > > > > **Response to Reviewer mniG on  "Clarification of our question"  Part (1/2)**
> > > > > > > > >
> > > > > > > > > Dear Reviewer mniG,
> > > > > > > > >
> > > > > > > > > Thanks so much for further clarification, which indeed helps us better understand your comment.
> > > > > > > > >
> > > > > > > > >
> > > > > > > > > **Q1: For flexibility, I mean the hyperparameter $k$ that determines the filter size is independent of the feature size $n$ --- one need not know $n$ beforehand when choosing $k$. For example, one can train a convolutional network on $n\times n$ inputs but test it for $n'\times n'$ (with $n \neq n'$). However, this proposed method does not have this flexibility.**
> > > > > > > > >
> > > > > > > > > A1: Thanks for clarifying.  We now understand the flexibility you define. In our method, the choice of $k$ does not rely on the value of $n$,  either. When $n$ changes, only the dilation changes and the size of the convolution kernel $k$ is unchanged.  We do not need to know $n$ to determine $k$.  Please refer to  Algorithm 1 in our manuscript. From line 1 to line 13, they are all independent to n. Only line 14 is related with $n$ which only determines the dilation.  Considering the case when the testing size $n'$ is 2 times as the training size $n$, the dilation $d = n/k$ would be doubled in testing and $k$ is unchanged.
> > > > > > > > >
> > > > > > > > >
> > > > > > > > > **Q2: Though not directly related to my comment, I would like to point out a misunderstanding --- the authors confuse the filter with the algorithm to perform filtering. In digital signal processing, the filter in BCOP is finite impulse response (FIR) (with finite receptive field). In contrast, the filter in CayleyConv/SOC is an infinite impulse response (IIR) (with a possible infinite receptive field). Since the inputs are circular-padded (and thus have an infinite length in math), an IIR filter applies on the infinite-dimensional space, where all input elements contribute to each output element. However, the notion of filter shall not be confused with the algorithm that performs the filtering operation. For example, one can also use FFT to compute a standard convolutional layer with kernel size  $k\times k$ -- the algorithm first zero-pads the kernel to $n\times n$, performs  Fourier transform on $n\times n$, etc. However, it does not make sense to say the algorithm increases kernel size from $k\times k$ to $n\times n$.**
> > > > > > > > >
> > > > > > > > >
> > > > > > > > > A2: Thanks again for your further clarification, we totally agree with your comments on FIR and IIR. In fact, we are not confused with an algorithm that performs the filtering operation. We also understand the example you give. In your example, the weight of a $n\times n$ convolution, $\mathbf{W}$, is zero-padded from a $k\times k$ kernel,  we can compute the convolution on the input $\mathbf{X}$ in spatial domain or frequency domain (they are equivalent).
> > > > > > > > >
> > > > > > > > >
> > > > > > > > > Frequency-domain formulation:
> > > > > > > > > \begin{equation}
> > > > > > > > > \mathbf{Y}=\mathrm{FFT}^{-1}(\mathrm{FFT}(\mathbf{W})\mathrm{FFT}(\mathbf{X})).
> > > > > > > > > \end{equation}
> > > > > > > > >
> > > > > > > > > Spatial-domain formulation:
> > > > > > > > > \begin{equation}
> > > > > > > > > \mathbf{Y}=\mathbf{W} \star \mathbf{X},
> > > > > > > > > \end{equation}
> > > > > > > > > where $ \star $ denotes the convolution operation.
> > > > > > > > >
> > > > > > > > > Since $ \mathbf{W}$ contains only $k\times k \times c_{in} \times c_{out}$ non-zero elements, of course, the reception field is always $k\times k$ no matter in spatial-domain formulation or frequency-domain formulation.
> > > > > > > > >
> > > > > > > > > However, in CayleyConv, the story changes. Below we explain why the reception field increases from $k\times k$ to $n\times n$ in detail.
> > > > > > > > >
> > > > > > > > > The frequency-domain formulation of CayleyConv is
> > > > > > > > > \begin{equation}
> > > > > > > > > \mathbf{Z}=\mathrm{FFT}^{-1}( C(\mathrm{FFT}(\mathbf{W})) \mathrm{FFT}({\mathbf{X}})),
> > > > > > > > > \end{equation}
> > > > > > > > > where $C(\mathrm{FFT}({\mathbf{W}}))=(\mathbf{I}-\mathrm{FFT}({\mathbf{W})+\mathrm{FFT}(\mathbf{W})^*})(\mathbf{I} +  \mathrm{FFT}({\mathbf{W}})-\mathrm{FFT}({\mathbf{W})^*})^{-1}$.
> > > > > > > > >
> > > > > > > > > Its equivalent spatial-domain formulation is
> > > > > > > > > \begin{equation}
> > > > > > > > > \mathbf{Z}=\mathrm{FFT}^{-1}(C(\mathrm{FFT}(\mathbf{W}))\star\mathbf{X},
> > > > > > > > > \end{equation}
> > > > > > > > >
> > > > > > > > > That is, the factual convolution kernel  of CayleyConv is $\mathrm{FFT}^{-1}(C(\mathrm{FFT}(\mathbf{W}))$ instead of $\mathbf{W}$.
> > > > > > > > >
> > > > > > > > > To know the receptive field size of $\mathrm{FFT}^{-1}(C(\mathrm{FFT}(\mathbf{W}))$, we can count the non-zeros in $\mathrm{FFT}^{-1}(C(\mathrm{FFT}(\mathbf{W}))$. Since $C(\mathrm{FFT}(\mathbf{W}))$ is not a period signal, the number of non-zero elements in $\mathrm{FFT}^{-1}(C(\mathrm{FFT}(\mathbf{W}))$ should be $n \times n\times c_{in}\times c_{out}$. That is, its receptive field size is $n\times n$. Even if $\mathbf{W}$ contains only $k\times k \times c_{in}\times c_{out}$ non-zeros, the factual convolution kernel  $\mathrm{FFT}^{-1}(C(\mathrm{FFT}(\mathbf{W}))$ contains $n\times n\times c_{in} \times c_{out}$ non-zeros. Thus, it increases the  size of receptive field from $k\times k$ to $n \times n$. Notably, if the Cayley function $C(\cdot)$ is replaced by indentity function $I(\cdot)$  ($I(\mathbf{X})=\mathbf{X}$),   $\mathrm{FFT}^{-1}(I(\mathrm{FFT}(\mathbf{W}))=\mathrm{FFT}^{-1}(\mathrm{FFT}(\mathbf{W})= \mathbf{W}$ (which is standard convolution) will contain only $k\times k\times c_{in}\times c_{out}$ non-zeros and the receptive field is still $k\times k$.

---

> > > > > > > > > > ### Comment · Reviewer_mniG · 2021-12-09
> > > > > > > > > > **Response to the flexibility of kernel size**
> > > > > > > > > >
> > > > > > > > > > I have double-checked the paper and the code. It seems there is a fundamental mistake, where the kernel size is $k = 3$ following the SOC paper. However, the paper only proves that the layer is orthogonal if the feature size $n$ is divisible by the kernel size $k$. Since the feature size $n$ is not divisible by the kernel size $3$, none of the convolutional layers is orthogonal.
> > > > > > > > > >
> > > > > > > > > > This fundamental mistake makes all experiments wrong, so I have to lower my score.

---

> > > > > > > > > > > ### Author Response · Authors · 2021-12-09
> > > > > > > > > > > **Response to "Response to the flexibility of kernel size"**
> > > > > > > > > > >
> > > > > > > > > > > Dear Reviewer mniG,
> > > > > > > > > > >
> > > > > > > > > > > Thanks for further checking.
> > > > > > > > > > > As we mentioned previously, if the feature size $n$ is not divisible by the kernel size $k$, we can add zero paddings to make the feature size divisible by $k$. To be specific, we need to pad $\lceil n/k \rceil *k - n $ rows/columns and the size of the  feature  is then $\lceil n/k \rceil *k$.
> > > > > > > > > > > That is why we set the dilation as  $\lceil n/k \rceil$ as mentioned in the last sentence of  the caption of Table 1. In this case, the convolution is still  orthogonal.
> > > > > > > > > > >
> > > > > > > > > > > Best,
> > > > > > > > > > >
> > > > > > > > > > > Authors

---

> > > > > > > > > ### Author Response · Authors · 2021-11-30
> > > > > > > > > **Response to Reviewer mniG on "Clarification of our question" Part (2/2)**
> > > > > > > > >
> > > > > > > > > Based on the above analysis on A2 to Q2, let us go back to Q1 to check the flexibility defined by the author that "one can train a convolutional network on $n\times n$ input but test it for $m\times m$ input (with $m \neq n$)" for CayleyConv.
> > > > > > > > >
> > > > > > > > > Again, let us recall that, the factual convolution kernel for CayleyConv is  $\mathrm{FFT}^{-1}(C(\mathrm{FFT}(\mathbf{W}))$, where $\mathbf{W}$ is obtained from zero-padding a $k\times k$ kernel by padding $n-k$ rows/columns, and $C(\mathrm{FFT}({\mathbf{W}})) = (\mathbf{I} -  \mathrm{FFT}({\mathbf{W}) + \mathrm{FFT}(\mathbf{W})^*} ) (\mathbf{I} +  \mathrm{FFT}({\mathbf{W}}) - \mathrm{FFT}({\mathbf{W})^*})^{-1}$.
> > > > > > > > >
> > > > > > > > > We denote  $\mathbf{W}$ for the training input size $n$ by $\mathbf{W}_{n}$ and denote that for the testing input size $m$ by  $\mathbf{W}_m$.  In this case,  $\mathbf{W}_n$ contains $n-k$ zero-pad rows/columns and $\mathbf{W}_m$ contains $m-k$ zero-pad rows/columns.
> > > > > > > > >
> > > > > > > > >  $\mathbf{W}_{n}$  and $\mathbf{W}_m$ contain the same non-zero elements, the only difference between them are the number of zero elements.
> > > > > > > > >
> > > > > > > > >
> > > > > > > > > However, the elements in  factual convolution kernel for CayleyConv $\mathrm{FFT}^{-1}(C(\mathrm{FFT}(\mathbf{W_{n}}))$ are quite different from $\mathrm{FFT}^{-1}(C(\mathrm{FFT}(\mathbf{W_{m}}))$ due to different number of zero paddings. Thus, for CayleyConv, a convolutional network  trained on $n\times n$ input might not work well for $m \times m$ input.
> > > > > > > > >
> > > > > > > > >
> > > > > > > > > More generally, we can use a unified formulation to include both standard convolution and CayleyConv by
> > > > > > > > > \begin{equation}
> > > > > > > > > \mathbf{Z} =  \mathrm{FFT}^{-1}( f(\mathrm{FFT}(\mathbf{W}) \mathrm{FFT}(\mathbf{X}))  =  \mathrm{FFT}^{-1}(f(\mathrm{FFT}(\mathbf{W})) \star \mathbf{X},
> > > > > > > > > \end{equation}
> > > > > > > > > where $\star$ denotes the convolution operation.
> > > > > > > > >
> > > > > > > > > When $f( )$ is an identity function, that is, $f(\mathbf{X}) = \mathbf{X}$, it will be standard convolution. When  $f(\mathbf{X})  = C(\mathbf{X})$, it will be CayleyConv.  The Cayley Transform $C()$ indeed changes the story a lot.  When $f( )$ is an identity function, the trained convolution kernel can well adapt the input of any size since the non-zero elements are unchanged as the input size change. However, when $f( )$ is Cayley Transform, the elements in factual convolution kernels   $\mathrm{FFT}^{-1}(C(\mathrm{FFT}(\mathbf{W}))$ change significantly as the number of zero-padding changes caused by the change of input size. In this case,  when the input size is $m \times m$ in the testing phase, the output from the convolution pre-trained on $n\times n$ might be useless for the final image recognition.

---

> > > > > > > > > ### Author Response · Authors · 2021-12-02
> > > > > > > > > **Additional Response to Reviewer mniG on "Clarification of our question"**
> > > > > > > > >
> > > > > > > > > Dear Reviewer mniG,
> > > > > > > > >
> > > > > > > > > We thank you again for your helpful comments which motivate us to think deeply about the limitations of our method and the previous methods.  Meanwhile,  we also deeply appreciate your patience in further clarifying your concerns and giving us more opportunities for clarification. This is a kind inquiry that if our clarification is clear? Do we still misunderstand your question?
> > > > > > > > > Please feel free to correct us if we still do not fully get your point, and we are still active in clarifying the issues.
> > > > > > > > >
> > > > > > > > > Best Regards,
> > > > > > > > >
> > > > > > > > > Authors

---

> ### Author Response · Authors · 2021-11-19
> **The second response to Reviewer mniG**
>
> Dear Reviewer,
> We thank you again for the helpful and inspiring feedback. Below we make more clarifications and we look forward to more feedback from you to help us improve the quality of the paper.
>
> **Q6:  (Casado & Martínez-Rubio 2019) provides an exact and efficient algorithm to parameterize orthogonal/unitary matrices. Using the current method, the authors will have to analyze how truncation error affects the exact orthogonality of the convolution kernel.**
>
> A6:  In the pointed work (Casado & Martínez-Rubio 2019), the weight orthogonalization is achieved through matrix exponential. To compute the exact matrix exponential in (Casado & Martínez-Rubio 2019), we need SVD, which is not friendly for parallelism and thus is slow in GPU planform.   To demonstrate the inefficiency issue in the exact matrix exponential, in the revised manuscript, we add Table 7  in Appendix E,  which directly compares the GPU latency of the exact matrix exponential and the truncated Taylor expansion. As shown in Table 7 (below table), the  truncated Taylor expansion is much faster than the exact matrix exponential.
>
> | matrix exponential | truncated Taylor expansion |
> |--------------------|----------------------------|
> | 12.8ms             | 2.5ms                      |
>
> Meanwhile, we add  Table 8 in Appendix E to show the truncated error between the truncated Taylor expansion $\sum_{0}^T \frac{\mathbf{A}^i}{i!}$ and the strict matrix exponential $\mathrm{exp}(\mathbf{A})$. Meanwhile, we also show the influence of the truncation errors on the orthogonality of the proposed ECO convolution. As shown in Table 8 (below table),  when $T=5$ in truncated Taylor expansion, our ECO convolution has achieved a very small truncation error and also obtains a marginal deviation from the strict orthogonal convolution.
>
> | T                | 1   |  2     | 3       | 4      | 5      | 10     |
> |------------------|------|-------|---------|--------|--------|--------|
> | $\|\|\mathrm{exp}(\mathbf{A}) - \sum_{0}^T \frac{\mathbf{A}^i}{i!}\|\|_2$| 0.48 | 0.16  | 0.04    | 0.008  | 0.001  | 0.0001 |
> |    $\frac{\|conv(\mathbf{x})\|_2}{\|\|\mathbf{x} \|\|_2}-1$               | 0.12 | 0.019 | 0.00038 | 0.0004 | 0.0000 | 0.0000 |
>
> **Q7: The matrix P is the Fourier transform of the convolution kernel, which generally is complex-valued. Restricting to the real-valued matrix has two drawbacks. a) The filters must be even, i.e., the method cannot construct orthogonal convolutions that are asymmetric. b) The space of real orthogonal matrices is disconnected, and the matrix exponential can only cover one component SO(c). I don't see why the authors do not adopt unitary matrices, which do not suffer from these two drawbacks.**
>
> A7:  Below we cite the statement in the original submitted manuscript for clarification (this paragraph is located above Table 1):
>
>
> In the current configuration, we set $\mathcal{P}_0[p,q,:,:]$ to be real matrix. In fact, $\mathcal{P}_0[p,q,:,:]$ can also be complex matrix.   When considering $\mathcal{P}_0[p,q,:,:]$ as a complex matrix, it makes the constructed orthogonal convolution kernel
>  ${\mathcal{W}}_0$ more complete. Meanwhile, it brings more computational cost in training. In Table 5 from Appendix  B of the originally submitted manuscript,  we have already compared the mode using a real $\mathcal{P}_0[p,q,:,:]$ with that using a complex $\mathcal{P}_0[p,q,:,:]$.
>
> Below we also attach the results shown in Table 5 from Appendix  B of the originally submitted manuscript to help you clearly see the comparisons between the real setting and the complex setting:
>
> |                | standard    | accuracy     | robust       | accuracy      | training     | time     |
> |------------------|------|-------|---------|--------|--------|--------|
> |                | LC-10    | LC-15     | LC-10       | LC-15      | LC-10    | LC-15     |
> | real | 73.64    | 60.20  | 74.72  | 63.99 | 37.1s|50.4s |
> |  complex                | 75.40 | 64.50 | 76.53 | 69.02 | 100.5s |117.3s |
>
> As shown in  the table, the complex setting achieves higher standard and robust accuracy since is it more complete. Meanwhile, the training time per epoch when using complex settings is around $2$ to $3$ times as that using the real setting.

---

> ### Author Response · Authors · 2021-11-29
> **A kind reminder for reviewer mniG  about the deadline of the last stage**
>
> Dear Reviewer mniG,
>
> This is a kind reminder that the deadline of the last stage is soon approaching. After that, we will not be able to respond to you.
> Therefore, we just want to double-check with you if your concerns have been well addressed or not?
> We are still here to answer some new questions or clarify some unclear issues.
>
> Sincerely,
>
> Authors

---

### Official Review · Reviewer_htXp · 2021-11-29

**Correctness:** 3
**Technical Novelty And Significance:** 3
**Empirical Novelty And Significance:** 3
**Recommendation:** 6
**Confidence:** 4

**Main Review:**

Benefits
-----------

- They show the link between singular vectors of the Jacobian of the convolutional layer and the convolutional kernel structure.
- The evaluation time of the proposed convolutional layer coincides with the evaluation time of the standard layer.
- They clearly describe how existing relative approaches differ from the proposed one.
- They obtain analytical formulae that yield a nice three-step approach to construct orthogonal convolution.

Drawbacks
---------------
- Longer training time for proposed ECO (Table 2)
- Simple datasets
- No black-box attacks


Suggestions
-----------------
- Add schematic visualization for forward and backward pass.
- Demonstrate results for more complex datasets (e.g. ImageNet)
- Add loss curves to see if there are effects aka standard&robust accuracy trade-off (similar to what is happening during adversarial learning)

**Summary Of The Paper:**

Summarizing the paper
-------------------------------

The paper proposes a method to construct a  convolutional layer with the orthogonal Jacobian matrix. Such a layer is 1-Lipschits: this property is an important one for building robust neural networks.
The authors conduct experiments on classification tasks using CIFAR10/CIFAR100 datasets. They report standard and robust accuracies, training, and evaluation time. They claim the proposed method outperforms existing state-of-the-art approaches.



**Summary Of The Review:**

Recommendation
------------------------
I recommend accepting the paper because the proposed method is new, addresses an important problem and empirically demonstrates its benefits.

---

> ### Author Response · Authors · 2021-11-29
> **Response to Reviewer htXp**
>
> Dear Reviewer htXp,
>
> We deeply appreciate that you value the novelty of the proposed method and acknowledge the importance of the solved problem, which is indeed encouraging for us.  Meanwhile, we thanks for your constructive suggestions on schematic visualization, more complex datasets and loss curves. These suggestions indeed help improve the quality of the submitted manuscript.  Since we are not allowed to revise the manuscript in current stage,  we will incorporate your suggestions when preparing the final version of manuscript.
>
>
> Sincerely,
>
> Authors

---

### Author Response · Authors · 2021-11-14
**Revision with New Experiments and Updated Texts based on Reviewers' Suggestions**

We thank all reviewers for their detailed reviews and valuable suggestions on improving our submitted manuscript. All reviewers commented positively on the novelty of the proposed method for constructing the orthogonal convolution in an explicit manner.  The main concerns are on the experiments.  In the revision, we have made the following modifications and we look forward to more feedback from the reviewers:


**Experiments**:

1)  According to the suggestions from Reviewer mniG,  In Appendix E, we explain why we adopt the truncated Taylor expansion to approximate the exact matrix exponential. Meanwhile, we add Table 8 to show the truncation error.  As shown in the table, the truncation error is very small, demonstrating the outstanding performance of the truncated Taylor expansion in approximating the exact matrix exponential.

2)   According to the suggestions from Reviewer 6wTV and YtM2, In Appendix C, we add Table 6 to show the performance of the standard convolution without orthogonal constraint. The results in Table 6 demonstrate the advantage of the proposed orthogonal convolution in the robust accuracy over the standard convolution.

3)  According to the suggestions from Reviewer YtM2,   in Appendix D, we add Table 7 to show the performance of the proposed ECO convolution based on the ResNet-18. The results in the table show that ours and other orthogonal convolution methods such as SOC and BCOP cannot achieve as high standard accuracy as the standard convolution when using the ResNet-18 backbone.  But the performance of orthogonal convolutions including SOC, BCOP and our ECO does not decrease significantly and is still acceptable.



**Texts**:
1) According to the suggestions from Reviewer mniG, at the end of the second paragraph of the Introduction (Section 1), we have changed the texts from "Straightforwardly, we can ..." to "It is plausibly straightforward to expand the convolution kernel to the doubly block circulant  Jacobian matrix and directly impose the orthogonal constraint on the Jacobian matrix. Nevertheless, it is extremely difficult to construct a Jacobian matrix that is both doubly block-circulant  and orthogonal."

2) According to the suggestions from 6wTV, we add the discussions on [1] orthogonal convolutional neural networks (CVPR 2020) and [2] deep isometric learning for visual recognition (ICML 2020) in the related work section. To be specific, [1,2] encourage the orthogonality of the Jacobian matrix of convolution layers through a regularizer, but they cannot achieve the strict orthogonality and  cannot attain the provable robustness.

3) Fix some typos. 1) change $\mathbf{F}_k$ to $\mathbf{F}_n $ in Eq. (3) based on the feedback from  Reviewer YtM2. 2) Fix the paragraph under Table 3 to make it align with the caption of Table 3 based on the feedback from  Reviewer YtM2.

---

### Author Response · Authors · 2021-11-26
**A kind reminder that the deadline of the second stage is approaching**

Dear Area Chairs and Reviewers,

Thanks again for your kind help in improving the quality of the submitted manuscript.
This is a kind reminder that the deadline of the second stage is approaching.
Please feel free to raise your question to the unclear points and we are very pleased to clarify the issues.
Hopefully, with our joint efforts, the manuscript can reach the level of acceptance.

Sincerely,

Authors

---

### Decision · Program_Chairs · 2022-01-20

**Decision:**

Accept (Poster)

**Comment:**

The paper studies the problem of how to construct orthogonal convolutional layers. It is known that a convolution layer is orthogonal if and only if its filters are obtained by certain Fourier operations on an orthogonal matrix. Previous work proposes to learn this orthogonal matrix, parameterized either through Cayley transform, or the exponential of a skew-symmetric matrix. This requires spectral computations with large matrices. The idea of this submission is to reduce the computational cost associated with this construction by letting this “core” matrix P be a periodic extension of a smaller orthogonal matrix P_0. Because of cancelations in the inverse DFT, this leads to sparse filters which can be implemented by dilated convolution.

The review process generated a very detailed discussion between authors and reviewers, with several important clarifications. Reviewers generally found that the paper contributes a novel construction of orthogonal convolution layers, with better efficiency at test time. Remaining concerns held by some reviewers include the limitations vis previous constructions of orthogonal convolution layers, questions about the efficacy of use of a Taylor expansion, certain minor limitations of the experiments. After detailed interaction with the authors, the reviewers converged to a decision to accept, motivated by the novelties of the construction and its advantages for test-time efficiency.